EMBO
Molecular Medicine

# TUBB1 mutations cause thyroid dysgenesis associated with abnormal platelet physiology

Athanasia Stoupa[1,2,3,4], Frédéric Adam[5], Dulanjalee Kariyawasam[3,4], Catherine Strassel[6], Sanjay Gawade[7], Gabor Szinnai[7,8], Alexandre Kauskot[5], Dominique Lasne[5,9], Carsten Janke[10,11] (iD), Kathiresan Natarajan[10,11,†], Alain Schmitt[1], Christine Bole-Feysot[12], Patrick Nitschke[13], Juliane Léger[3,14,15,16], Fabienne Jabot-Hanin[13], Frédéric Tores[13], Anita Michel[6], Arnold Munnich[17,18], Claude Besmond[17], Raphaël Scharfmann[1], François Lanza[6], Delphine Borgel[5,9], Michel Polak[1,2,3,4,19] & Aurore Carré[1,2,3,*] (iD)

## Abstract

The genetic causes of congenital hypothyroidism due to thyroid dysgenesis (TD) remain largely unknown. We identified three novel *TUBB1* gene mutations that co-segregated with TD in three distinct families leading to 1.1% of *TUBB1* mutations in TD study cohort. *TUBB1* (Tubulin, Beta 1 Class VI) encodes for a member of the β-tubulin protein family. *TUBB1* gene is expressed in the developing and adult thyroid in humans and mice. All three *TUBB1* mutations lead to non-functional α/β-tubulin dimers that cannot be incorporated into microtubules. In mice, *Tubb1* knock-out disrupted microtubule integrity by preventing β1-tubulin incorporation and impaired thyroid migration and thyroid hormone secretion. In addition, *TUBB1* mutations caused the formation of macroplatelets and hyperaggregation of human platelets after stimulation by low doses of agonists. Our data highlight unexpected roles for β1-tubulin in thyroid development and in platelet physiology. Finally, these findings expand the spectrum of the rare paediatric diseases related to mutations in tubulin-coding genes and provide new insights into the genetic background and mechanisms involved in congenital hypothyroidism and thyroid dysgenesis.

**Keywords** congenital hypothyroidism; macroplatelets; mutations; thyroid dysgenesis; TUBB1
**Subject Categories** Genetics, Gene Therapy & Genetic Disease; Haematology

## Introduction

Thyroid dysgenesis (TD) is a feature in 65% of patients with congenital hypothyroidism (CH), the most common neonatal endocrine disorder affecting one in 2,500–3,500 newborns (Deladoëy *et al*, 2011; Barry *et al*, 2016). In France, the prevalence of CH due to TD is estimated in 1/5,000 (Barry *et al*, 2016). TD includes a vast spectrum of developmental thyroid anomalies encompassing athyreosis, thyroid ectopia, hypoplasia of an orthotopic gland, and hemithyroid (Barry *et al*, 2016; Stoupa *et al*, 2016). During embryogenesis, the midline thyroid anlage appears on embryonic day E.8.5 in mice and at 3 gestational weeks (GW) in humans. The midline anlage and ultimobranchial bodies migrate and fuse in the definitive

1   INSERM U1016, Faculté de Médecine, Cochin Institute, Université Paris Descartes, Sorbonne Paris Cité, Paris, France
2   IMAGINE Institute Affiliate, Paris, France
3   RARE Disorder Center: Centre des Maladies Endocriniennes Rares de la Croissance et du Développement, Paris, France
4   Pediatric Endocrinology, Gynecology and Diabetology Unit, Hôpital Universitaire Necker-Enfants Malades, AP-HP, Paris, France
5   INSERM UMR_S1176, Paris-Sud University, Université Paris-Saclay, Le Kremlin-Bicêtre, France
6   INSERM, EFS Grand Est, BPPS UMR-S 1255, FMTS, Université de Strasbourg, Strasbourg, France
7   Department of Biomedicine, Pediatric Immunology, University of Basel, Basel, Switzerland
8   Pediatric Endocrinology, University Children's Hospital Basel, University of Basel, Basel, Switzerland
9   Necker Children's Hospital, Biological Hematology Service, Assistance Publique—Hôpitaux de Paris, Paris, France
10  Institut Curie, CNRS UMR3348, PSL Research University, Orsay, France
11  Institut Curie, CNRS UMR3348, Université Paris Sud, Université Paris-Saclay, Orsay, France
12  Genomic Platform, INSERM UMR 1163, IMAGINE Institute, Paris Descartes University, Sorbonne Paris Cité, Paris, France
13  Bioinformatics Platform, IMAGINE Institute, Paris Descartes University, Paris, France
14  Pediatric Endocrinology Unit, Hôpital Universitaire Robert Debré, AP-HP, Paris, France
15  Paris Diderot University, Sorbonne Paris Cité, Paris, France
16  INSERM UMR 1141, DHU Protect, Paris, France
17  INSERM U1163, IMAGINE Institute, Translational Genetics, Université Paris Descartes, Sorbonne Paris Cité, Paris, France
18  Department of Genetics, Hôpital Universitaire Necker-Enfants Malades, AP-HP, Paris, France
19  Fédération Parisienne pour le Dépistage et la Prévention des Handicaps de l'Enfant (FPDPHE), Paris, France
*Corresponding author. Tel: +33 1 76 53 55 71; E-mail: aurore.carre@inserm.fr
†Present address: Physiology and Biomedical Engineering Department, Mayo Clinic, Rochester, MN, USA

pretracheal position on E13.5 in mice and at 7 GW in humans (Trueba et al, 2005; Nilsson & Fagman, 2013). The cells differentiate into thyrocytes organized in follicles or C cells (Appendix Table S1; Szinnai et al, 2007). Abnormalities at any step of thyroid development may result in TD associated with hypothyroidism or not (Maiorana et al, 2003). Previous studies of sporadic and familial TD covering a wide clinical spectrum identified mutations in nine genes: PAX8, NKX2-1, FOXE1, NKX2-5, TSHR, GLIS3, NTN1, JAG1 and BOREALIN (Dentice et al, 2006; Senée et al, 2006; Carré et al, 2009, 2014, 2017; Sura-Trueba et al, 2009; Ramos et al, 2014; Opitz et al, 2015; de Filippis et al, 2016). However, mutations in these genes are found in only 5% of all patients with TD and identification of causative mutations remains a challenging task. Our objectives were to extend the knowledge on the genetic basis of CH and TD. We performed WES (whole exome sequencing; Choi et al, 2009; Hildebrandt et al, 2009) for siblings with CH and thereby identified a mutation in TUBB1 gene. Then, we analysed 270 TD cases by targeted NGS including the TUBB1 gene. We identified two more TUBB1 mutations in patients with CH and TD.

TUBB1 (Tubulin, Beta 1 Class VI) encodes for a member of the β-tubulin protein family. β-tubulins are one of two core protein families that heterodimerize to form α/β-tubulin dimers, which assemble into microtubules, one of the major cytoskeletal structures. The β1 isotype of tubulin (TUBB1) has been described as specifically expressed in platelets and megakaryocytes and involved in proplatelet formation and platelet release (Patel et al, 2015). Few mutations of TUBB1 have been identified in patients with a rare autosomal dominant disease congenital macrothrombocytopaenia, in which impaired microtubule assembly results in low platelet counts and macroplatelets (Kunishima et al, 2009; 2014; Bastida et al, 2018; Johnson et al, 2016; Burley et al, 2018). Tubb1-knock-out mice have thrombocytopaenia and spherical platelets (Schwer et al, 2001), but not known thyroid abnormalities. Our results highlight a hitherto unsuspected role for a specific tubulin isotype, Tubb1, in thyroid development and disease and extend our knowledge on genetic background of CH.

# Results

### Identification of TUBB1 mutations in a family with TD

Family F1 is a consanguineous family of Algerian descent. The parents are first cousins (I.1, I.2) with five children including two females [II.1 (patient P1) and II.2 (patient P2)] with CH. Both patients were born at full term and diagnosed with CH by routine neonatal screening (Fig 1), which showed thyroid-stimulating hormone (TSH) elevation (164 and 177 μIU/ml in P1 and P2, respectively). On days 13 and 11, TSH was 67 and 202 μIU/ml in P1 and P2, respectively (normal range, N: 0.3–7 μIU/ml), and free thyroxine (T4) was 14 and 13.3 pmol in P1 and P2, respectively (N: 9.5–25 pmol; Fig 1). L-thyroxine therapy was initiated. [123]I scintigraphy showed thyroid ectopia in both siblings. Another sibling (II.5, P3), aged 12 years, had thyroid hypoplasia (thyroid volume, 3.1 ml; N: 7 ± 3 ml) with a small right pyramidal lobe (17 *2 mm) and normal thyroid function tests. The parents had normal thyroid function, and two other siblings (II.3 and II.4) had normal thyroid function but were not able to undergo thyroid ultrasonography.

To look for genetic causes of CH in P1 and P2, we performed whole exome sequencing (WES) using the variant filtering and prioritization strategy described in Appendix Fig S1. Using the recessive transmission model, WES identified a novel missense homozygous TUBB1 mutation (c.479C>T, p.P160L, rs759117911) in both siblings with CH (P1 and P2) and in the sibling with thyroid hypoplasia (P3; Fig 1). Both parents and sibling II.3 were carriers. The remaining sibling (II.4) did not carry the mutation. WES identified no variants in genes known to be associated with TD or thyroid dyshormonogenesis.

### Search for TUBB1 mutations in a cohort with thyroid dysgenesis (TD) and congenital hypothyroidism (CH)

After identification of the above-described novel TUBB1 mutation, we used targeted next-generation sequencing (NGS) to assess TUBB1 in a cohort of 270 patients with CH and TD. In a second family (F2) with a father (I.2) of Moroccan and a mother (I.1) of French descent, a female with CH and thyroid gland ectopia (P4, II.1) had a heterozygous TUBB1 mutation (c.318C>G, p.Y106X; Fig 1). CH was diagnosed upon routine neonatal screening (TSH, 250 μIU/ml) and confirmed on day 15 (TSH, 1,100 μIU/ml; free T4, 3.5 pmol/l; and free T3, 2.45 pmol/l). Thyroid scintigraphy showed an ectopic thyroid. The father carried the same heterozygous mutation; unfortunately, thyroid ultrasound was not performed, and complete phenotype was therefore not possible. In a paternal aunt (I.3, P5), an evaluation at 26 years of age for obesity and depression showed mild hypothyroidism (TSH, 6.6 μIU/ml; N: 0.1–5.5 μIU/ml; free T4, 8.7 pmol/l; N: 9.8–23.1 pmol/l). Thyroid ultrasonography and scintigraphy showed right hemithyroid.

In a third family (F3), a patient (II.1, P6) with CH and an ectopic thyroid was shown by targeted NGS to have a heterozygous frameshift TUBB1 mutation (c.35delG, p.Cys12Leufs*12, rs77324804) that created a premature stop codon at amino acid 23 (Fig 1). CH was diagnosed neonatally based on TSH elevation (476 μIU/ml) and low free T4 and T3 levels (8 and 5.6 pmol/l, respectively). L-thyroxine therapy was started at 11 days of age. Both parents were of French descent. The father (I.1, P7) carried the same heterozygous mutation and had normal thyroid function with mild thyroid lobe asymmetry by ultrasonography (right lobe, 6.9 ml; left lobe, 5 ml). The other siblings and mother had normal thyroid function and morphology and did not carry the mutation.

By targeted NGS, neither P4 nor P6 had any variants in genes known to cause CH (with TD or dyshormonogenesis).

In the Exome Aggregation Consortium (ExAC) database, estimated allele frequencies are 0.000008 for the c.479C>T mutation (rs759117911, 20:57598961 C/T) and 0.000025 for the c35delG (rs773248042, 20:57594611 TG/T) mutation. Neither mutation has been reported in homozygous form. The c.318C>G variation has not been reported in public databases. The in silico prediction tools, PolyPhen-2, SIFT, predict that c.479C>T is probably damaging and deleterious, respectively, with a PHRED-scaled CADD score of 32 (damaging: > 15; Kircher et al, 2014). The other two mutations create a premature stop codon and have PHRED-scaled CADD score of 35. CADD is a prediction algorithm, which integrates information contained in more than 60 diverse annotations of genetic variation into a single score. The higher the CADD score, the higher the deleteriousness probability of the variant investigated.

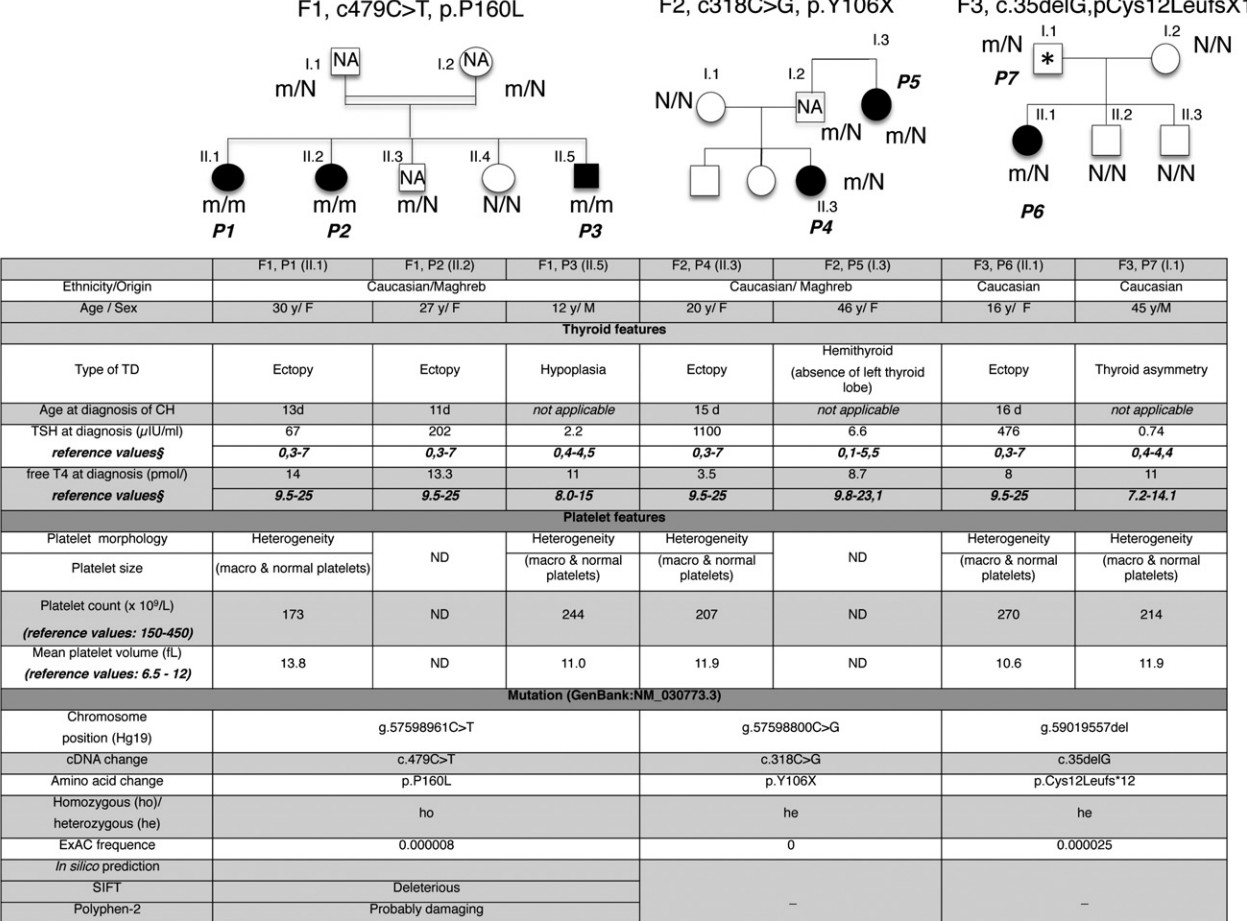

**Figure 1. Pedigrees and clinical table.**

Pedigrees of three families with *TUBB1* mutations. Family F1 has three affected individuals with homozygous mutations, family F2 has two affected individuals with heterozygous mutations, and family F3 has two affected individuals with heterozygous mutations. Thus, all seven patients (P1–P7) carry at least one mutated allele and have thyroid dysgenesis (TD) and macroplatelets. The patients are represented with filled symbols and unaffected family members with open symbols. NA, not available thyroid ultrasonography; *mild thyroid asymmetry (possibly normal) with normal thyroid function; N, not mutated; m, mutated.

A Burden test was applied to determine whether *TUBB1* was significantly enriched in rare variants in the 270 patients with CH and TD versus 406 Caucasian controls from the 1000 Genomes project. The patients in the TD cohort had athyreosis, ectopia, hemithyroid or hypoplasia. In the TD group, 14/270 patients (5.2%) exhibited at least one rare functional variant in *TUBB1* compared with 8/406 controls (2%). This difference is significant from a statistical point of view ($P = 0.0227$). None of the patients in our cohort with *TUBB1* mutations had athyreosis. Performing the same test after excluding the 77 patients with athyreosis increased the significance of the difference of TD group ($n = 193$) versus controls ($P = 0.0095$). The list of rare functional variants found in *TUBB1* is provided in Table EV1.

The three amino acids affected by the *TUBB1* mutations are strictly conserved across species, from humans to zebrafish, and across all β-tubulins (Fig 2A). All three mutations were located in the first part of *TUBB1*, i.e., in the N-terminal domain needed for guanosine triphosphate (GTP) activity (Fig 2B). The c.318C>G and

c.35delG mutations created a premature stop codon, thereby removing the intermediate and C-terminal domains required for microtubule-associated protein (MAP) binding (Nogales *et al*, 1998).

In sum, we identified three *TUBB1* mutations in three independent families of patients with CH and TD chiefly manifesting as thyroid gland ectopia. Thus, we found 1.1% of *TUBB1* mutations in patients affected with CH and TD in our cohort.

**β1-tubulin is expressed in the developing thyroid in humans and mice**

β1-tubulin expression has so far been reported only in megakaryocytes and platelets (Wang *et al*, 1986; Lecine *et al*, 2000). Our finding of *TUBB1* mutations in patients with TD prompted us to look for β1-tubulin expression in thyroid tissue. In human thyroid tissue, *TUBB1* mRNA was expressed at 8, 10 and 12 GW and in adulthood (Fig 3A). In mouse thyroid tissue, *Tubb1* was expressed at E13.5 and strongly at E15.5, E17.5 and adulthood (Fig 3B). To refine our

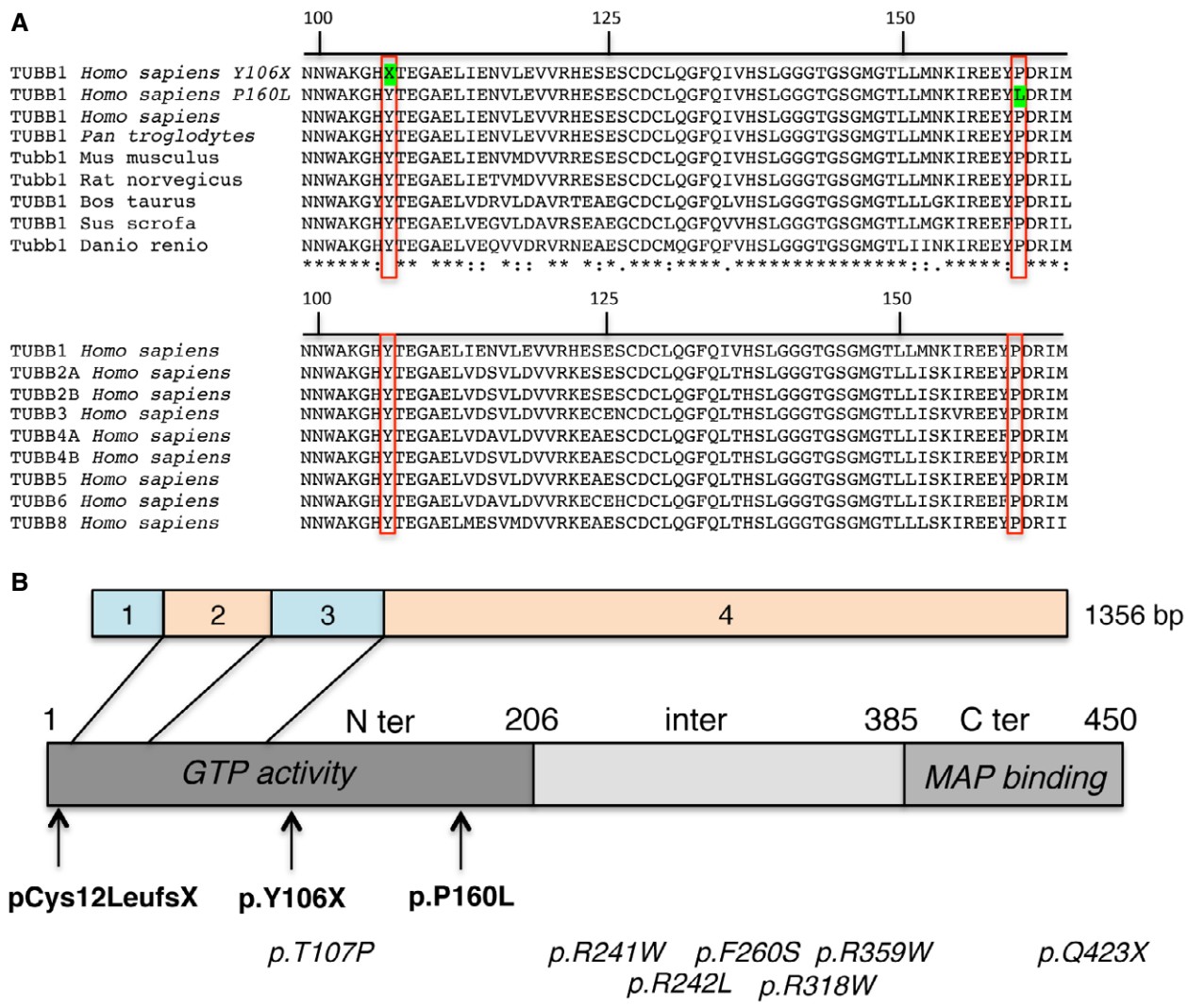

**Figure 2. Molecular genetics.**

A Alignment and conservation of residues encoded by *TUBB1* orthologues and genes encoding for other β-tubulins (*TUBB2A, TUBB2B, TUBB3, TUBB4A, TUBB4B, TUBB5, TUBB6* and *TUBB8*). Mutations are shown in green. Mutated aminoacids through species or through all TUBB are shown in red boxes.

B Location of *TUBB1* mutations in the cDNA and of the corresponding changes in the protein. Exons are represented by boxes numbered from 1 to 4. The dark grey box represents the protein domain responsible for encoding guanosine triphosphate (GTP) and the light grey box the domain for microtubule-associated protein (MAP) binding. The arrows show the consequences of the three *TUBB1* mutations in our patients, all of which are in the GTP domain. Mutations of this study are in bold, and published mutations associated with congenital macrothrombocytopaenia are in italic.

study of *Tubb1* expression, we used cells sorted from mice thyroid tissue based on well-accepted markers (Gawade *et al*, 2016) with stringent sorting regions, including the brightest cells for each selected marker (Pecam for endothelial cells, EpCAM for epithelial cells, Pdgfra for fibroblasts and CD45 for leucocytes). As expected, expression was strongest in platelets sorted using the specific megakaryocyte lineage marker CD41 (Appendix Fig S2A). However, *Tubb1* was also expressed in EpCAM-positive epithelial-cell populations containing thyrocytes, at E17.5 and in adulthood (Fig 3C).

Similarly, in human thyroid tissue, immunohistochemistry showed β1-tubulin expression in the cytoplasm of thyroglobulin (TG)-producing thyrocytes at 12 GW (Fig 3D). Comparable findings were obtained with mouse thyroid tissue (Appendix Fig S2B). No staining was observed in the thyroid tissue of $Tubb1^{-/-}$ mice

(Appendix Fig S2C). These data established that β1-tubulin is expressed in thyrocytes.

**Functional *in vitro* analysis of disease-causing mutations**

To further investigate the implication of *TUBB1* gene mutations in thyroid disease, we transfected the mutations into the Nthy-ori 3-1 cell line. Only the c479C>T (p.P160L) mutation could be studied in this model. The c318C>G (p.Y106X) and c.35delG (p.Cys12Leufs*12) mutations created premature stop codons that yielded truncated proteins (Appendix Fig S3) that cannot form functional α/β-tubulin dimers and thus cannot get incorporated into microtubules (Nogales *et al*, 1998; Joe *et al*, 2009). The pathogenic role of these mutations is thus loss-of-function.

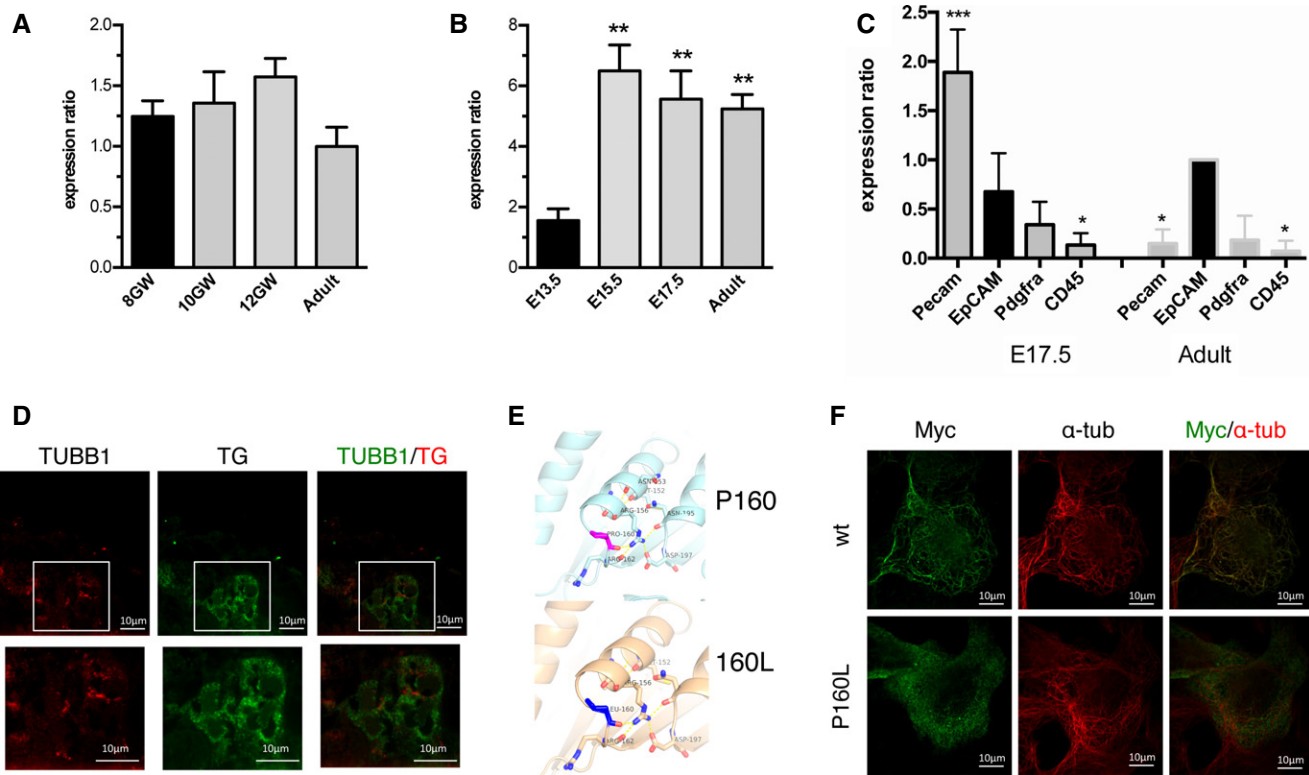

**Figure 3.  β1-tubulin expression in the developing thyroid and deleterious effect of *TUBB1* mutations.**

A   *TUBB1* is expressed in the developing human thyroid at 8 gestational weeks (GW), 10 GW and 12 GW and in the adult human thyroid: quantitative PCR results normalized to one of three thyroid tissues at 8 GW value (in black) and peptidylprolyl isomerase A. Experiments with four tissues per stage. Results are reported as mean ± SEM. Statistical comparisons versus 8 GW (in black) using ANOVA test showed no significant differences.

B   *Tubb1* is expressed in the developing mouse thyroid at E13.5, E15.5 and E17.5 and in the adult mouse thyroid: quantitative PCR results normalized to one of three thyroid tissues at E13.5 value and peptidylprolyl isomerase A. Experiments with four tissues per stage. Results are reported as mean ± SEM. Statistical comparisons versus E13.5 (black), $**P < 0.01$ using ANOVA test.

C   Sorted mouse thyroid cells, *Tubb1* expression in endothelial cells (Pecam-positive cells), epithelial cells (EpCAM), fibroblasts (Pdgfra) and leucocytes (CD45) at E17.5 and adulthood: quantitative PCR results (normalized to EpCAM-positive cells at adult stage and peptidylprolyl isomerase A). Mean ± SEM of three independent cell-sorting experiments. Statistical comparisons versus epithelial cells (in black) at each stage, $*P < 0.05$, $***P < 0.001$ using ANOVA test.

D   Immunohistofluorescence of β1-tubulin (in red), thyroglobulin (TG, in green) and both merged (from left to right) at 12 GW in human thyroid tissue. The boxes delineate the enlarged areas shown below.

E   Structural representation of wild-type and p.P160L mutated β1-tubulin showing the change in conformation surrounding helix H4. Top: proline at position 160 in pink; bottom: substituting leucine at the same position in blue.

F   Localization of transfected Myc-tagged wild-type and mutant (P160L) β1-tubulin in Nthy cells. Immunostaining with anti-Myc antibody (in green) and microtubule cytoskeleton stained with α-tubulin antibody (in red). Note that β1-tubulin -P160L is found throughout the cytoplasm showing puncta appearance, whereas wild-type β1-tubulin colocalizes with the microtubules.

*Structural modelling of p.P160L mutant β1-tubulin*

To assess the consequences of the P160L mutation on β1-tubulin function, the sole missense mutation that should give rise to a full-length protein, we first compared the configurations of the mutated and wild-type proteins. The P160L mutation is located at the end of helix H4 (Fig 3E). In wild-type β1-tubulin, the proline residue stabilizes loop H4-S5 and places R156 in a position that promotes a salt bridge interaction with D197 in the β-sheet S6. Furthermore, R162 in loop H4-S5 and N195 in the β-sheet S6 establish hydrogen bonds with R156. Mutation of proline 160 to leucine might affect the loop conformation of the H4-S5 loop and thus disrupting the interaction network mediated by R156. Hence, the P160L mutation is most likely affecting the conformation of the β-tubulin, which could lead to dysfunctions of the α/β-tubulin dimer.

*In vitro consequences of the p.P160L β1-tubulin mutation on microtubule incorporation*

Following the predictions from the modelling, we examined the ability of β1-tubulin P160L protein to incorporate within the microtubule network in the Nthy cell line. We used a vector containing human TUBB1 cDNA tagged with Myc. We transfected Nthy cells with the wild-type and P160L mutant then compared the distribution and location of the protein by double-label immunofluorescence (Fig 3F). The overexpressed wild-type β1-tubulin co-assembled incorporates into microtubules, in contrast to β1-tubulin P160L (absence of incorporation). Thus, the P160L mutation clearly affects the capacity of β1-tubulin to incorporate into microtubules, which is most likely related to defects at the structural level, thus forming a dysfunctional α/β-tubulin dimer.

In summary, all three *TUBB1* mutations identified here likely lead to non-functional α/β-tubulin dimers that cannot be incorporated into microtubules.

### Tubb1 knock-out in mice affects thyroid development and function

The phenotype of patients bearing *TUBB1* mutations suggests that β1-tubulin may contribute to thyroid development and function. We assessed this hypothesis in *Tubb1* knock-out mice (*Tubb1*$^{-/-}$; Schwer *et al*, 2001).

We found significant increases in expression levels of the other β-tubulin isoforms (Tubb2a, Tubb5, Tubb2b and Tubb3) in E17.5 thyroids of *Tubb1*$^{-/-}$ mice compared to wild-type mice (Appendix Fig S4A). Tubb2b and Tubb3 expression levels were also increased in adult *Tubb1*$^{-/-}$ mice. In addition, Tuba3 and Tuba4 expression in *Tubb1*$^{-/-}$ mouse thyroids were diminished at E17.5 and in adults. Interestingly, these compensatory changes in expression levels seen in the thyroid gland mirror those described in platelets of *Tubb1*$^{-/-}$ mice (Appendix Fig S4A; Schwer *et al*, 2001).

### Thyroid gland development and differentiation

**Thyroid gland morphology in mice**    To study thyroid gland development from the early stages of budding and migration of the median anlage (MA) and ultimobranchial bodies (UB) to the late stages of differentiation, we used immunohistochemistry, surface quantification and quantitative PCR (Appendix Table S1). For most of the experiments, we used Nkx2-1 as a marker of progenitor and differentiated thyroid cells. At E9.5, thyroid anlage surface area and proliferation ratio were significantly greater in *Tubb1*$^{-/-}$ mice than in wild-type mice (Fig 4A and B, and Appendix Table S2). At E11.5, thyroid migration was slightly delayed in three of six *Tubb1*$^{-/-}$ embryos, a few Nkx2-1-positive cells were visible along the migration tract of the mutants, and thyroid surface area was not different between mutant and wild-type embryos (Fig 4A and B, and Appendix Table S2). During late thyroid development, at E13.5, fusion of the median anlage and ultimobranchial bodies was slightly delayed in three of four *Tubb1*$^{-/-}$ embryos (Fig 4A and Appendix Table S2). After E13.5, *Tubb1*$^{-/-}$ thyroids were significantly hypoplastic (Fig 4A and B). Moreover, at E17.5, a supplementary pyramidal lobe was visible near the normal lobe in two of six embryos (Fig 4A). In keeping with this finding, patient P3 in family 1 described above, who was homozygous for the p.P160L mutation, had a hypoplastic thyroid with a pyramidal lobe. In sum, the thyroid phenotype of *Tubb1*$^{-/-}$ mice indicates that β1-tubulin is required for normal thyroid migration and morphology.

**Thyroid gene ontogeny in mice**    We used quantitative PCR to assess the ontogeny of genes involved in thyroid function and development (Fig 4C). Expression levels of mRNAs for thyroglobulin (Tg), thyroid peroxidase (Tpo), TSH receptor (Tshr) and calcitonin (Calca), late differentiation markers, were significantly decreased at E15.5, as were those of Tg, Tpo and Calca at E17.5, in *Tubb1*$^{-/-}$ compared to wild-type embryos. Significant decreases were also observed at E17.5 in the *Tubb1*$^{-/-}$ embryos in the early thyroid development markers Nkx2-1, Pax8 and Foxe1. These results indicate impaired thyroid gland development and differentiation in *Tubb1*$^{-/-}$ mice.

**Endocrine signature at completion of thyroid gland development**    T4-positive thyroid surface area relative to total thyroid surface area was shown by immunohistochemistry to be significantly increased at E17.5 in *Tubb1*$^{-/-}$ versus wild-type embryos (Fig 4D). Calcitonin-positive thyroid surface area relative to total thyroid surface area tended to be greater in the mutants ($P = 0.19$). Thus, final thyroid differentiation was abnormal in *Tubb1*$^{-/-}$ embryos, with increases in intrathyroidal T4 that probably reflected impaired hormone secretion.

These results establish a role for β1-tubulin in thyroid development and differentiation.

### Thyroid gland function and structure in adult mice

We compared thyroid hormone status in adult *Tubb1*$^{-/-}$ and wild-type mice. At 3 months of age, serum TSH levels were higher and T4 levels lower in *Tubb1*$^{-/-}$ than in wild-type mice, suggesting hypothyroidism in the mutants (Fig 5A). These results are consistent with the finding of CH in our patients carrying *TUBB1* mutations.

Furthermore, we examined thyroid structure in adult mice (3 months of age). Surface area was not different between the *Tubb1*$^{-/-}$ and wild-type thyroids (data not shown). Moreover, the thyroid tissue was disorganized in the mutants with large regions without organized follicles (Fig 5B). When we used electron microscopy to examine thyrocyte ultrastructure (Fig 5C), we found larger numbers of dense and rod-shaped vesicles in *Tubb1*$^{-/-}$ compared to wild-type thyrocytes. These data were consistent with impaired T4 release responsible for hypothyroidism. Marked endoplasmic reticulum (ER) dilation was seen in *Tubb1*$^{-/-}$ compared to wild-type thyrocytes (Fig 5C), indicating ER stress, which was confirmed by the findings of increased *Chop* and *XBPs* expression by quantitative PCR and of increased Chop protein levels by Western blotting in *Tubb1*$^{-/-}$ thyroids (Fig 5D). By immunohistochemistry, the ER marker KDEL co-localized with TG in *Tubb1*$^{-/-}$ and wild-type mice but was especially abundant in disorganized areas of adult *Tubb1*$^{-/-}$ thyroids (Fig 5E).

These data indicated partial thyroid tissue disorganization, vesicle accumulation and ER stress in *Tubb1*$^{-/-}$ mice.

In sum, our findings in *Tubb1*$^{-/-}$ mice demonstrate a complex mechanism of hypothyroidism involving early abnormal proliferation of progenitors, delayed thyroid migration, defective thyroid differentiation, impaired thyroid hormone release and structural disorganization of the thyroid tissue. The abnormal thyroid migration may explain the thyroid phenotype found in patients carrying *TUBB1* mutations, especially those with thyroid gland ectopia.

### Analysis of platelets from patients bearing *TUBB1* mutations

Until now, β1-tubulin expression had been reported only in the megakaryocyte lineage (Patel *et al*, 2015). *TUBB1* mutations have been previously reported to cause macrothrombocytopaenia (Kunishima *et al*, 2009, 2014; Fiore *et al*, 2017). We therefore studied the platelets of the above-described patients with *TUBB1* mutations and thyroid gland abnormalities.

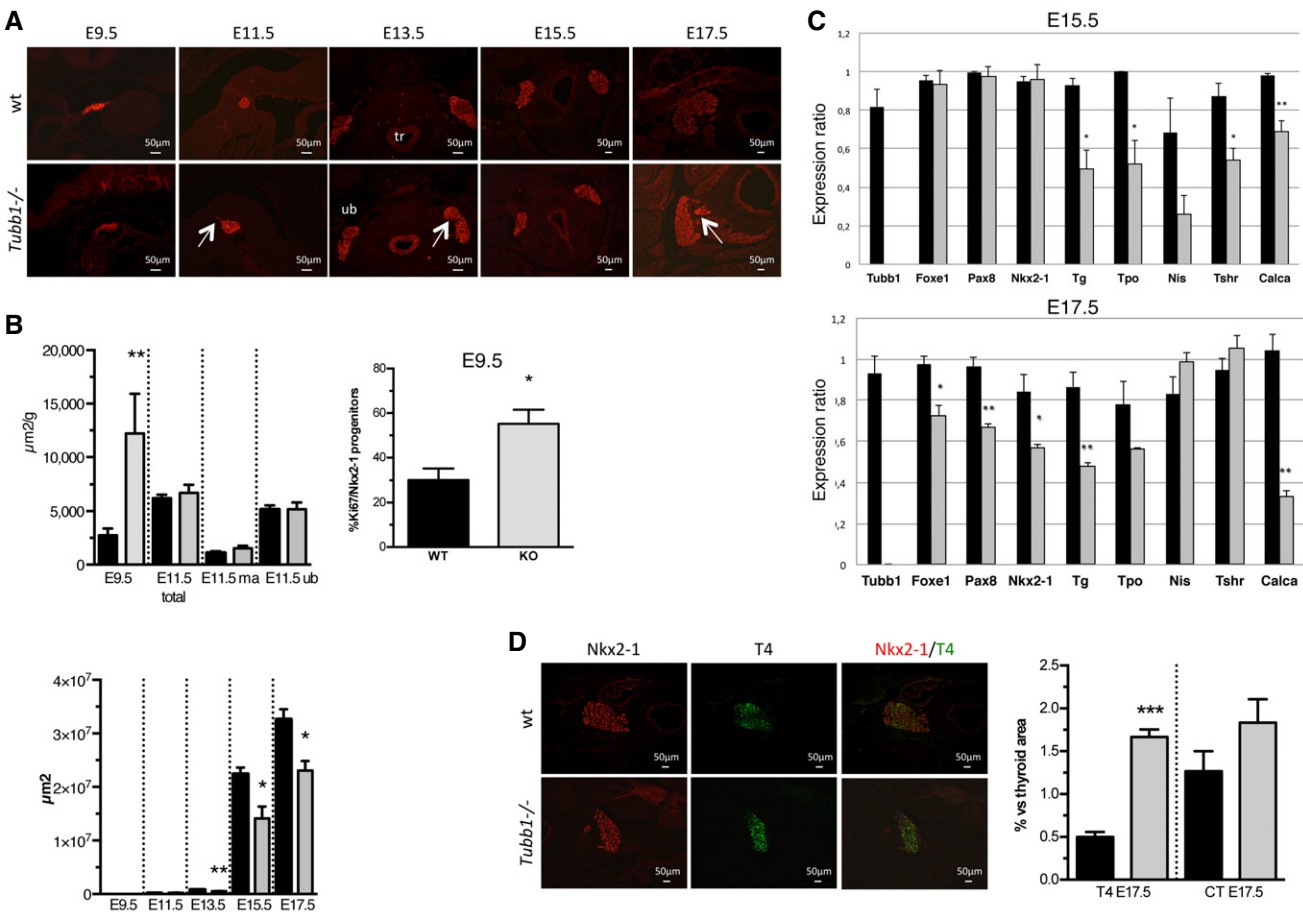

**Figure 4.** $Tubb1^{-/-}$ **mice have abnormal thyroid development.**

A   Thyroid morphology was investigated using Nkx2-1 staining at E9.5 and E11.5 (sagittal sections) and at E13.5, E15.5 and E17.5 (transverse sections) in wild-type (wt) and $Tubb1^{-/-}$ littermates. Nkx2-1 was used as a marker of early thyrocyte progenitors and differentiated thyrocytes. Delays in thyroid migration and in fusion of the median anlage to the ultimobranchial bodies were observed at E11.5 and E13.5 (arrow), respectively, in $Tubb1^{-/-}$ mice. At E11.5, stained migrating thyroid cells were visible along the tract only in $Tubb1^{-/-}$ littermates (arrow). At E17.5, a pyramidal lobe was visible in $Tubb1^{-/-}$ littermates (arrow). tr: trachea; ub: ultimobranchial body. Scale bar: 50 μm.

B   Total thyroid surface area (μm²) was quantified using Nkx2-1 staining at each embryonic stage. Top: thyroid surface area normalized for weight of each embryo. Four mice were analysed per genotype at E9.5 and five mice at E11.5. Bottom: thyroid surface area in μm². Four mice were analysed per genotype at E9.5, five at E11.5, six at E13.5, four at E15.5 and three at E17.5. Right: proliferation ratio calculated as the proportion of Nkx2-1-positive cells labelled with Ki67 at E9.5 in wt and $Tubb1^{-/-}$ thyroid anlages. Four mice were analysed for wild-type genotype and five for $Tubb1^{-/-}$. ma: median anlage. Note that the thyroid ma is larger and the proliferation ratio higher at E9.5 in $Tubb1^{-/-}$ mice, whereas from E13.5 onwards the $Tubb1^{-/-}$ thyroids were hypoplastic. Wt in black, KO, $Tubb1^{-/-}$ in grey.

C   Thyroid marker expression by quantitative PCR in thyroid tissue at E15.5 and E17.5 in $Tubb1^{-/-}$ versus wt mice normalized for peptidylprolyl isomerase A. Note the decreases in thyroid differentiation markers at E15.5 and E17.5 and also in thyroid transcription factors at E17.5. Four mice were analysed per genotype at E15.5 and three at E17.5.

D   Staining of Nkx2-1 and T4 at E17.5 and surface area quantification demonstrating T4 retention in $Tubb1^{-/-}$ versus wt thyroids. The data are the percentage of T4 or calcitonin (CT) surface area versus total thyroid area (estimated from the Nkx2-1-stained surface area). Three mice were analysed per genotype. Scale bar: 50 μm.

Data information: Results are reported as mean ± SEM. Student's *t*-test, *$P < 0.05$, **$P < 0.01$ and ***$P < 0.001$.

---

### Haematological parameters (Fig 1)

Thrombocytopaenia was a feature in patients with *TUBB1* mutations studied by other groups (Kunishima *et al*, 2009, 2014; Fiore *et al*, 2017). However, when we used an automated haematology analyser to measure the platelet count and mean platelet volume (MPV) in our seven patients, we consistently found normal platelet counts. In contrast, MPV was above the normal range in P1 and near the upper limit of normal in P4 and P7. A blood smear analysis (Fig 6A) demonstrated variations in platelet size with the presence of macro-platelets. Moreover, parents I.1 and I.2 of family F1 had MPV values

at the top of the normal range (10.2 and 11.3 fl, respectively; normal, 7.5–11.2 fl) with many macroplatelets. Electron microscopy confirmed the large platelet size (Fig 6B).

The increased platelet size could be related to defects in their biogenesis. To study this, we cultured megakaryocytes from patients P1 and P4 from peripheral blood progenitors and analysed proplatelet formation (Fig 6C). Interestingly, shaft thickness and coiled element diameter of the future platelets were significantly increased compared to controls, indicating that the *TUBB1* mutations affected proplatelet formation and platelet size.

     

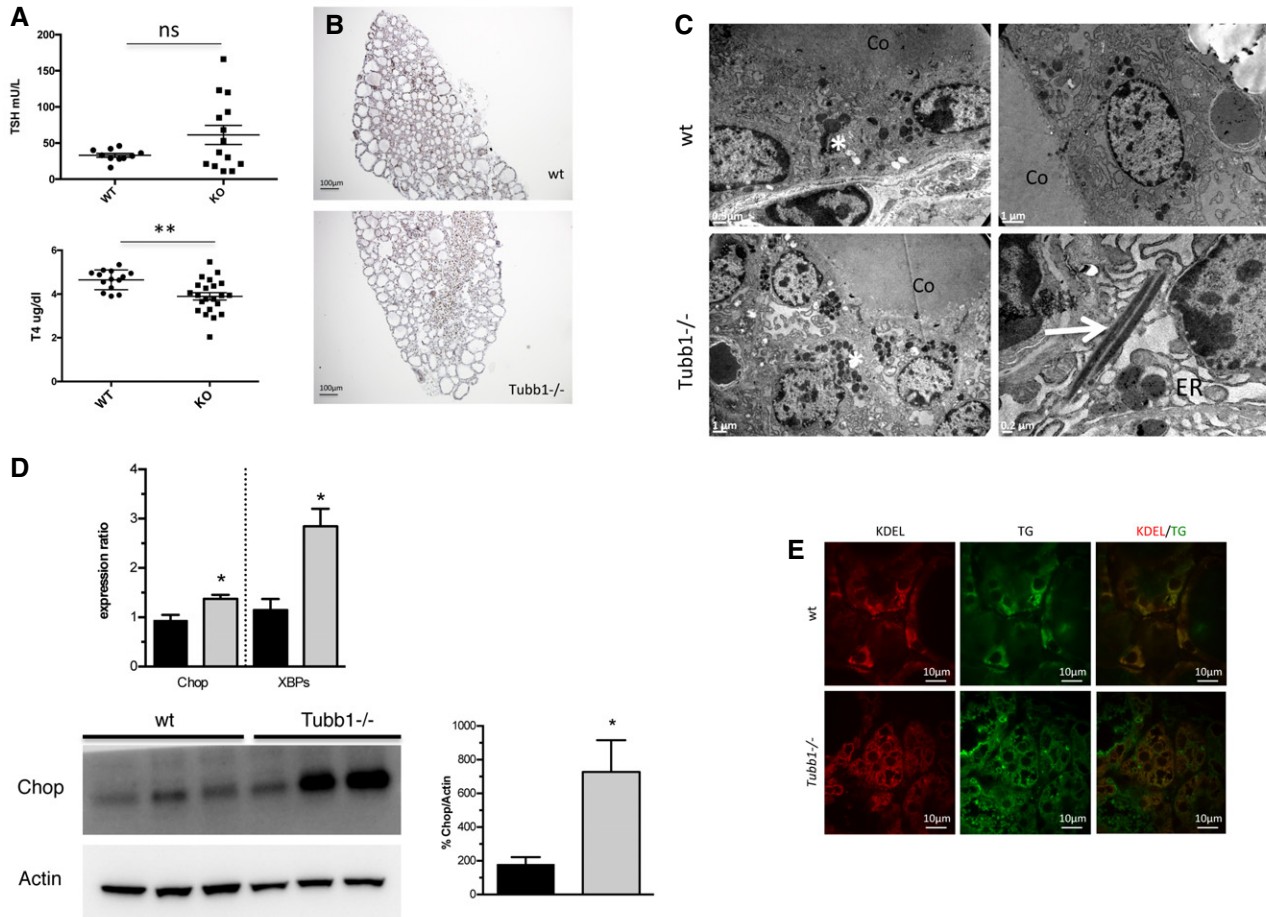

**Figure 5.** *Tubb1⁻/⁻* mice have altered thyroid function with impaired hormone secretion.

A   Serum TSH and T4 levels in 3-month-old *Tubb1⁻/⁻* and wild-type mice. Numbers of animals tested were 10 wild-type males and 14 *Tubb1⁻/⁻* males for TSH and for 14 wild-type and 22 *Tubb1⁻/⁻* males for T4. *Tubb1⁻/⁻* mice had hypothyroidism with elevated TSH and decreased T4 versus wild-type mice.

B   Nkx2-1 (in brown) immunostaining of adult thyroid tissue. Note the disorganization of the thyroid tissue in *Tubb1⁻/⁻* versus wild-type mice.

C   Ultrastructural alterations shown by electron microscopy in wild-type and *Tubb1⁻/⁻* thyroid tissues. In *Tubb1⁻/⁻* tissues, note the disorganization of secretion vesicles (white asterisks) and rods of identical density to secretion vesicles (white arrow). The ER is considerably dilated in *Tubb1⁻/⁻* thyrocytes. Representative views. Scale bars at the bottom left for each view. Co: colloid.

D   ER stress in *Tubb1⁻/⁻* thyroid tissue. Top: Chop and XBPs expression by quantitative PCR in adults, normalized to peptidylprolyl isomerase A. Experiments with four tissues per stage for each genotype; wt in black and *Tubb1⁻/⁻* in grey. Bottom: Chop and Actin protein expression by Western blotting in three representative wild-type and *Tubb1⁻/⁻* mice. The Chop band quantification normalized for Actin confirms the increased Chop expression demonstrated by quantitative PCR in *Tubb1⁻/⁻* versus wild-type thyroids. All lanes are from the same blot, which was cut where indicated.

E   Co-immunostaining of endoplasmic reticulum (ER) marker (KDEL, in red), thyroglobulin (Tg in green) and both merged (from left to right) in adult thyroid tissue. Note the thyrocyte disorganization in *Tubb1⁻/⁻* mice.

Data information: Results are reported as mean ± SEM. Student's *t*-test, *$P < 0.05$, **$P < 0.01$.
Source data are available online for this figure.

---

*Functional analysis of human platelets*

We first quantified β1-tubulin expression in the platelets of our patients and found significant decreases ($P < 0.001$) of $45.0 \pm 3.7\%$ and $36.5 \pm 3.4\%$ in P1 and P3 (F1), respectively; $45.0 \pm 3.8\%$ in P4 (F2); and $29.1 \pm 3.3\%$ and $30.6 \pm 6.2\%$ in P6 and P7 (F3), respectively. Expression of α-tubulin was normal (Fig 7A). The β1-tubulin antibody used in our study detects only the C-terminal part of the protein. Consequently, in the patients of families F2 and F3, whose mutations created premature stop codons, only the wild-type protein was detected. In the F1 patients, who were homozygous for the mutation, the results suggested either diminished protein expression or protein instability.

Then, to investigate whether the *TUBB1* mutations affected platelet function, we assessed platelet activation by flow cytometry using a specific monoclonal antibody (PAC1), which recognized the active conformation of the integrin α$_{IIb}$β$_3$, a platelet activation marker. PAC1 binding was significantly increased in P1 and P3, indicating abnormal platelet activation, whereas in P6 and P7 (F3) α$_{IIb}$β$_3$ activations were comparable to those in controls (Fig 7B). We were unable to evaluate P4 (F2).

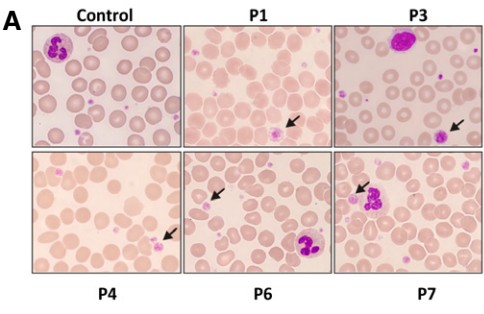

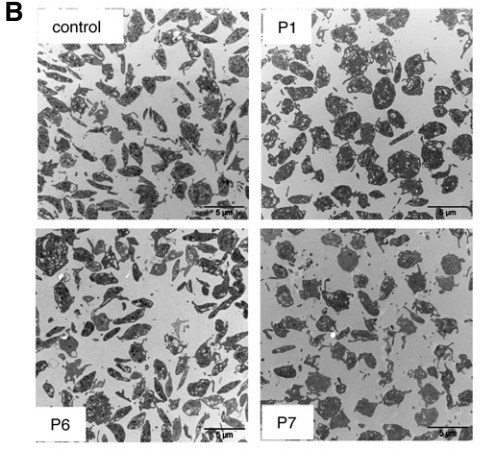

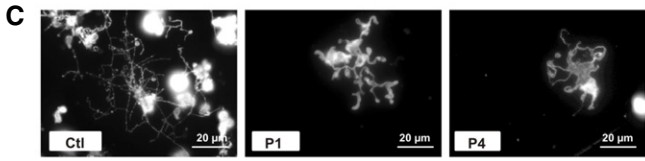

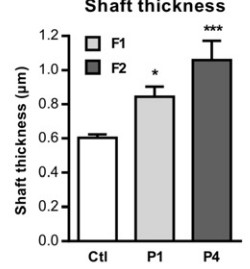

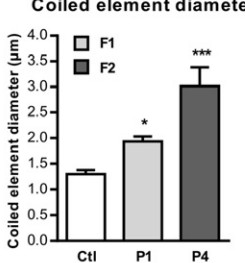

**Figure 6.  *TUBB1* mutations induce abnormal platelet formation or macroplatelets.**

A   Representative blood smears of patients with *TUBB1* mutations (P1 and P3 homozygous for p.P160L, P4 heterozygous for p.Y106X, and P6 and P7 heterozygous for c.35delG) after May–Grünwald–Giemsa staining. Variable platelet size in patients was observed with all three mutations with the presence of macroplatelets (arrows).

B   Transmission electron microscopy (TEM) analysis of platelet ultrastructure in the patients. The results confirm the abnormal platelet morphology.

C   Representative images of proplatelet formation in P1 and P4. Cultured control and patient megakaryocytes (MKs) after thrombopoietin-induced differentiation were spread over a BSA-coated coverslip on day 10. On day 13 or 14, MKs were fixed (4% paraformaldehyde) and proplatelet structure was observed after β-tubulin staining. Scale bars: 20 μm. Only P1 and P4 could be studied. The graphs show the mean ± SEM of the shaft thickness of proplatelet extensions (left) and the diameter of coiled elements (right) of the future proplatelets, both of which were significantly increased in the patients, indicating that the *TUBB1* mutations (homozygous p.P160L and heterozygous p.Y106X) affected proplatelet morphology. Statistical significance was determined by one-way ANOVA, followed by Dunnett's multiple comparisons test (*$P < 0.05$, ***$P < 0.001$; 20–60 MKs were analysed/patient).

Finally, we investigated the aggregations of washed platelets upon activation by ADP or collagen, two key platelet agonists (Fig 7C). Aggregation in response to ADP was normal in P6 and P7 (F3) but increased in P1 and P3 (F1) and in P4 (F2) compared to controls. Similarly, collagen-induced platelet aggregation was normal in P6 and P7 but was increased in P1 and P4, even with low doses. These abnormal platelet aggregations were not related to hypothyroidism or patient's treatment by L-thyroxine, since they were not found in three patients with thyroid ectopy not harbouring *TUBB1* mutations and under the same treatment (Appendix Fig S5). Moreover, it should be noted that these results were confirmed by at least three independent investigations for P1 and P3 and two for P6. The other patients were studied once. None of the nine controls exhibited similar hyperaggregation profile.

In summary, these results indicate that p.P160L (F1), p.Y106X (F2) and c.35delG (F3) *TUBB1* mutations affect β1-tubulin expression in platelets and result in abnormally large platelet size, probably as a consequence of proplatelet abnormal formation. Moreover, the p.P160L mutation induces significant platelet activation in resting condition and hyperaggregation in response to agonists, whereas the c.35delG mutation does not seem to affect platelet function.

# Discussion

We identified three *TUBB1* mutations in patients with TD and macroplatelets. In a consanguineous family (F1), two females with CH and thyroid gland ectopia had the same homozygous *TUBB1* mutation, and their brother had thyroid gland hypoplasia with normal function. Of 270 patients with CH and TD, two probands from unrelated families had heterozygous *TUBB1* mutations. All patients with heterozygous or homozygous *TUBB1* mutations had TD and macroplatelets. Thyroid gland ectopia was the most common form of TD, but mild thyroid asymmetry, thyroid hypoplasia and hemithyroid were also seen.

This is the first time that CH with TD was associated with macroplatelets, and we demonstrate in our study that *TUBB1* mutations are the common cause. β1-tubulin expression has heretofore been

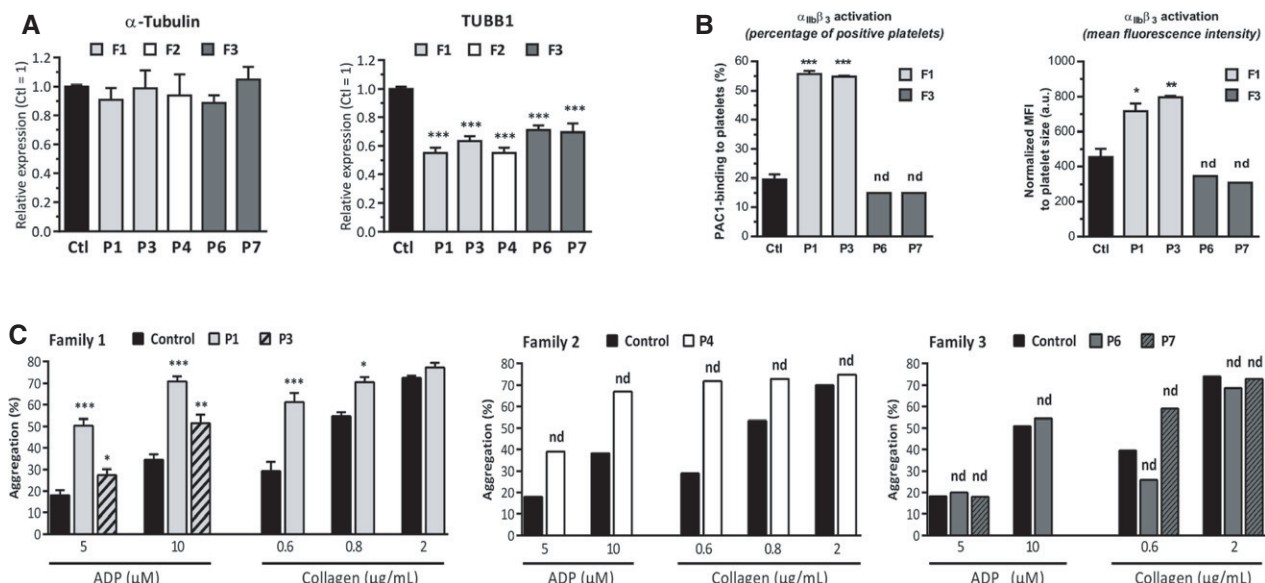

**Figure 7. TUBB1 mutations alter protein expression in platelets and platelet functions.**

A Relative expression of α-tubulin and β1-tubulin in platelets from P1 and P3 (homozygous p.P160L), P4 (heterozygous p.Y106X), and P6 and P7 (heterozygous c.35delG) was quantified by Western blotting using specific antibodies against α-tubulin or β1-tubulin. With both mutations, α-tubulin expression was normal but β1-tubulin expression was significantly decreased. The β1-tubulin antibody recognizes C-terminal domain.

B αIIbβ3 integrin activation was evaluated by flow cytometry (% of positive platelets and mean fluorescence intensity) in whole blood from P1 and P3 (homozygous p.P160L) and from P6 and P7 (heterozygous c.35delG); P4 could not be studied. While no αIIbβ3 integrin activations were detected in P6 and P7 (family 3), P1 and P3 (family 1) exhibited a significant increase in αIIbβ3 activation, indicating basal platelet activation.

C Aggregation of washed platelets induced by ADP (5 or 10 μM) or collagen (0.6, 0.8 or 2 μg/ml) was evaluated in at least one member of each family. Patients from families 1 and 2 had increased platelet aggregation in response to low doses of agonist versus controls, suggesting that TUBB1 affected by homozygous p.P160L and heterozygous p.Y106X mutations affected platelet function. Platelet aggregation to both agonists was normal in both patients from family 3 (heterozygous c.35delG).

Data information: Results are reported as mean ± SEM. Family F1, $n = 3$ independent experiments; Family F2, $n = 1$ experiment; Family F3, $n = 2$ independent experiments. Statistical significance was determined by one-way ANOVA, followed by Dunnett's multiple comparisons test (*$P < 0.05$, **$P < 0.01$, ***$P < 0.001$, nd: statistical significance not determined because $n \leq 2$).

described as confined to megakaryocytes and platelets. Our findings demonstrate that β1-tubulin is also expressed in developing and adult thyroid tissue. Furthermore, CH and TD were found in humans with *TUBB1* mutations and in *Tubb1*$^{-/-}$ mice. These data indicate that normal thyroid development and function require β1-tubulin incorporation into the microtubule network. Furthermore, our patients with *TUBB1* mutations had normal platelet counts but abnormal platelet morphology, and two of the three mutations (p.P160L and p.Y106X) were associated with abnormal platelet function. In contrast, in previous studies, humans with *TUBB1* mutations and *Tubb1*$^{-/-}$ mice had macrothrombocytopaenia but no reported thyroid disorders (Kunishima *et al*, 2009, 2014; Fiore *et al*, 2017).

The previously reported *TUBB1* mutations affect the intermediate or C-terminal domain of β1-tubulin, whereas the three novel mutations described here modify the N-terminal domain. However, during the drafting of this work, Bastida *et al* (2018) published a genetic analysis of a cohort of patients with inherited platelet disorders where they identified four novel *TUBB1* mutations: two in the C-terminal domain and two in the N-terminal domain, including the same mutation that we described here for family F3, c.35delG. In their study, the patient with the c.35delG mutation had a thrombocytopaenia, which differs from the normal platelet count of our family F3. This discrepancy could be explained by another factor, different from the *TUBB1* mutation, which could induce thrombocytopaenia.

Two of our described mutations gave stop codon earlier but not the p.P160L mutation. The p.P160L protein expression was decreased in the platelets of patients, but the hypothesis of the protein instability was not conclusive *in vitro* experiments (data not shown). Others mechanisms should be considered. Strikingly, all three mutations are loss-of-function mutations: two of the mutations lead to truncated β1-tubulin, which cannot form functional α/β-tubulin dimers and thus cannot be incorporated into microtubules. The third is a missense mutation (P160L), which, despite expressing full-length β-tubulin, still does not allow the formation of functional α/β-tubulin dimers to be integrated into the microtubules network. Hence, the three *TUBB1* mutations were deleterious for the β1-tubulin function, which is confirmed on the functional level, as their phenotypes are similar to those of the *Tubb1*$^{-/-}$ mouse. Consequently, *Tubb1*$^{-/-}$ mice emerged as a useful tool for studying the impact of impaired β1-tubulin function on the thyroid gland. Considering the hypothyroidism phenotype present only in *Nkx2-1*$^{-/-}$ or *Pax8*$^{-/-}$ or double *Nkx2-1*$^{+/-}$*Pax8*$^{+/-}$ mice, we decided to study directly the knock-out mice (Kimura *et al*, 1996; Mansouri *et al*, 1998; Amendola *et al*, 2005). Haploinsufficiency of one of these critical genes for thyroid development causes hypothyroidism in humans, whereas in mice, the homozygous deletion leads to thyroid phenotype.

Our experiments in *Tubb1*$^{-/-}$ mice provided the first evidence of a role for β1-tubulin in TD and hypothyroidism. The thyroid glands

of *Tubb1*$^{-/-}$ mice exhibited abnormal proliferation at E9.5, default of migration at E11.5 and E13.5, and hormone secretion failure at E17.5 and adulthood. All these mechanisms require normal microtubule organization and function. The idea that microtubule functions are fine-tuned and thus adapted to specific cellular role by the expression of specific tubulin isotypes is an emerging concept known as the "tubulin code" (Janke, 2014). Tubulin mutations have been more and more linked to different human disorders, in particular to neurological disorders (Chakraborti *et al*, 2016). Microtubules are essential for mitosis to unfold normally (Prosser & Pelletier, 2017). Here, we show that invalidation of β-tubulin isotype *Tubb1* perturbs mitosis of progenitors at E9.5, thus inducing hyperproliferation. Mitotic phenotypes have already described during neuronal development in mice with loss of another β-tubulin isotype, *Tubb5* (Breuss *et al*, 2012). Moreover, normal thyroid development requires that progenitor proliferation occurs during a specific time window (Nilsson & Fagman, 2017). Early proliferation at E9.5, as described by our group in *Hes1*$^{-/-}$ mice (Carre *et al*, 2011), results in abnormal thyroid development, probably via impairment of the pool of progenitors dedicated to thyroid development. Furthermore, microtubules are required for cell orientation during migration (Ladoux *et al*, 2016). Our data in the *Tubb1*$^{-/-}$ mice show a delay of thyroid progenitors cell migration during thyroid development, which again is mirrored by *Tubb5*$^{-/-}$ that show impaired neuronal migration (Breuss *et al*, 2012). In our patients with *TUBB1* mutations, thyroid gland ectopia was the most common form of TD (4/7 patients), further supporting a role for β1-tubulin in thyroid gland migration. The delay of thyroid bud migration could be reminiscent of the dual ectopy seen in 9% of CH due to ectopy (Wildi-Runge *et al*, 2012). Finally, endosome/lysosome trafficking is taking place on microtubules (Huotari & Helenius, 2011; Raiborg *et al*, 2015; Bonifacino & Neefjes, 2017). Endosome-to-lysosome transport of thyroid hormones via a vesicular transport system has been described in the thyroid gland (Rousset *et al*, 2015; Carvalho & Dupuy, 2017). A proper thyroid microtubule integrity is required for thyroid hormone secretion established by previous studies done in the 1970s (Nève *et al*, 1972; Wolff & Bhattacharyya, 1975). While so far nothing is known on the precise role of tubulin isotypes and microtubules organization in trafficking of thyroid hormone vesicles, there is strong evidence that transport along microtubules is affected by the tubulin code (Nirschl *et al*, 2016). Our data are the first to demonstrate that a specific tubulin isotype, β1-tubulin, is required for proper vesicle trafficking and thyroid hormone release into the bloodstream, and they extend our knowledge about the association between microtubules and secretion vesicles in the thyroid gland. Furthermore, we have shown an increased expression of ER stress markers in the *Tubb1*$^{-/-}$ thyroid. Previous studies demonstrated ER stress with activation of the unfolded protein response in association with reduced thyroglobulin (TG) synthesis and TG accumulation within the ER (Gaide Chevronnay *et al*, 2015; Di Jeso & Arvan, 2016). We can conclude that thyroid dysgenesis is a feature in *TUBB1* mutation carriers and in *Tubb1*$^{-/-}$ mice. The data on variable degrees of hypothyroidism in humans corroborate with the thyroid function data in mice as the variable TSH values have been found.

β1-tubulin is the most divergent isotype of tubulin, and its incorporation into microtubules is expected to change the properties of these filaments. Losing β1-tubulin thus certainly alters microtubule properties, and our data provide strong evidence that these particular properties are necessary for adequate thyroid development and function.

Our patients with *TUBB1* mutations had macroplatelets. This abnormal platelet size is further evidence that *TUBB1* mutations adversely affect platelet morphology probably as a consequence of proplatelets formation. In these previously described mutations, only the p.R318W mutation has been investigated in platelet aggregations and it did not induce defect of platelet function (Kunishima *et al*, 2009). For the other mutations (Kunishima *et al*, 2014; Johnson *et al*, 2016; Fiore *et al*, 2017; Bastida *et al*, 2018), no bleeding tendency have been reported. Here, our patients with p.P160L and p.Y106X mutations exhibited an unexpected increase in platelet functions, raising the question whether these platelet function abnormalities result in clinical symptoms such as thrombosis requires evaluation, particularly in older patients who may be at increased risk. Thyroid dysfunction can affect the haemostatic balance (Squizzato *et al*, 2007; Kyriakakis *et al*, 2016), with clinical effects that vary across thyroid disorders (Franchini *et al*, 2010). Moreover, MPV correlated positively with the TSH level (Kim *et al*, 2013). A *TUBB1* mutation screening study in patients with hypothyroidism and altered MPV and/or a history of thrombotic disease would be of great interest. Therefore, patients bearing a *TUBB1* mutation with cardiovascular risk factors could be monitored and the question arises whether antiplatelet drugs might be effective as prophylaxis (Kuhli-Hattenbach *et al*, 2017). Conversely, thyroid function and morphology assessment could be suggested in patients with congenital macrothrombocytopaenia.

The c.35delG mutation in P6 and P7 did not cause hyperaggregation. The patients with the c.35delG mutation had the same thyroid phenotype as other families with other mutations. At least one patient per family had thyroid gland ectopia with CH. Neither thyroid nor platelet phenotype severity correlated with the type of mutation. Further investigations are needed to elucidate the difference between the effects of c.35delG and p.P160L/p.Y106X on platelet function.

The common feature in *TUBB1* mutation carriers is thyroid dysgenesis and abnormal platelet morphology. *TUBB1* mutations constitute a model of dominant inheritance of CH with TD. Most known mutations responsible for TD and previously described *TUBB1* mutations causing macrothrombocytopaenia are also dominant (Kunishima *et al*, 2009, 2014; Guillot *et al*, 2010; Ramos *et al*, 2014; Carré *et al*, 2017; Fiore *et al*, 2017). Our data indicate high penetrance for platelet alterations and incomplete penetrance with variable expressivity for TD, ranging from TD without CH to TD with CH and leading to variable types of TD (ectopia, hypoplasia, hemithyroid or asymmetric thyroid gland). No patient had athyreosis. Indeed, in the described familial pedigrees, some carriers have mild or no thyroid phenotype, suggesting that the *TUBB1* germline mutation may be necessary to be affected by CHTD but it is probably not sufficient to display the phenotype. A second hit (such as a somatic mutation in the thyroid or an epigenetic defect) could be the additional prerequisite to express the disease. Furthermore, the hypothesis of random autosomal monoallelic expression in the thyroid could explain the difference in intrafamilial phenotypic variability in F3. These hypotheses are already documented in the literature for TD (Deladoëy *et al*, 2007; Magne *et al*, 2016). Finally, the genetics of TD remains complex with mutations in more than nine known genes and both classical and complex modes of inheritance, such as a suggested

oligogenic model by Persani et al (de Filippis et al, 2017). The same genetic pattern of inheritance is also observed in other endocrine-related disorders such as congenital hypogonadotropic hypogonadism (Boehm et al, 2015) or in the more complicated genetic model of Bardet–Biedl syndrome (Muller et al, 2010).

TUBB1 mutations were found in only 1.1% of our cohort with CH and TD patients. The Burden test showed enrichment in rare TUBB1 variants carriers in the cohort versus controls. Our data increment the number of predisposing genes for thyroid dysgenesis but with a novel phenotype associating platelet disorder.

Taken together, our data indicate heretofore unsuspected roles for a specific isotype of β-tubulin, Tubb1, in thyroid development and function, while also confirming its importance for microtubule integrity and platelet function. Loss-of-function of TUBB1 mutations impairs β1-tubulin incorporation into microtubules. Our results confirm that normal thyroid-cell proliferation and thyroid migration are essential to thyroid gland development. Thyroid hormone secretion requires β1-tubulin incorporation into the microtubules, suggesting a specific function of this tubulin isotype in intracellular transport of vesicles. Thus, our work provides novel insights into the role of the Tubb1 isotype in thyroid physiopathology and in platelet function and therefore expands the spectrum of the rare paediatric diseases related to tubulin mutations and microtubule dysfunction.

# Materials and Methods

### Patients

We have a large study cohort of patients with congenital hypothyroidism due to TD. Patients have been diagnosed with primary CH during the first days after birth by the systematic neonatal screening in France (TSH > 15 mU/l). Diagnosis of CH was confirmed with a control blood sample during the first weeks of life and imaging tests (thyroid ultrasonography and thyroid scintigraphy). A consanguineous family in which two children had TD and CH and another had small thyroid gland with normal thyroid function was initially studied. Subsequently, genetic testing was performed in 270 patients (184 girls and 86 boys) with CH and TD (ectopic thyroid gland, $n = 167$; athyreosis, $n = 77$; hemithyroid $n = 20$; and thyroid hypoplasia, $n = 6$). This study was approved by the review board (Ethics Committee, Ile de France, Paris, France: P11012-IDRCB 2012-A00797-36). The written informed consent forms were collected, and the experiments conformed to the principles set out in the WMA Declaration of Helsinki and the Department of Health and Human Services Belmont Report.

### Detection of mutations in humans

Genomic DNA was isolated from whole blood. Exome capture and sequencing were performed at the genomics platform of the IMAGINE Institute. WES libraries were prepared from 3 μg genomic DNA per individual, which was sheared by ultrasonication (Covaris S220 Ultrasonicator, Woburn, MA, USA). Exome capture was performed using the SureSelect Human All Exon V6 Kit (Agilent Technologies, Santa Clara, CA, USA). The resulting libraries were sequenced on a HiSeq 2500 HT device (Illumina, San Diego, CA, USA) according to the manufacturer's recommendations. Paired-end ($2 \times 130$) 76-bp reads were generated and mapped on the human

reference genome. More than 97% of the exome was covered at least 30 times. Raw data were analysed as described (Gordon et al, 2013), using an in-house software system (Polyquery). The variant prioritization strategy was as follows: (i) selection of functional (protein-altering) variants (removal of intergenic and 3′/5′ UTR variants, non-splice-related intronic variants and synonymous variants); (ii) variants with a frequency below 1% in public databases (dbSNP, 1000 Genomes, EVS, ExAC; release date, January 2018); and (iii) variants previously identified in fewer than five individuals contributing 11,811 in-house exomes (Appendix Fig S1).

The HypothySeq NGS Panel included 78 genes known to be associated with CH (TD; dyshormonogenesis; defects in thyroid hormone (TH) transport proteins; and inborn errors in TH membrane transport, metabolism or action) and candidate genes validated in animal models (mouse and zebrafish knock-out models) or by microarray assays but not yet validated in humans. This panel was previously validated using controls including samples from positive controls with known thyroid disease-causing mutations, to assess sensitivity (false-negative rate); and from healthy individuals screened by WES for another research study, to test specificity (false-positive rate). Genomic DNA libraries were created using SureSelectXT Target Enrichment Reagent Kit (Agilent Technologies) and subjected to custom-targeted DNA panel enrichment. In the 78 genes associated with CH, 1,006 regions of interest were captured by the corresponding 120-bp cRNA baits, using SureDesign software (Agilent Technologies; Homo sapiens, hg19, GRCh37, February 2009). The 233,103-bp targeted DNA regions (protein-coding exons of the main isoform and supplementary coding exons of each gene, including 25-bp flanking intronic sequences) were sequenced on Illumina HiSeq 2500 (Illumina). This step generated $2 \times 130$ paired-end reads. Bioinformatic analyses included alignment against the reference genome, variant calling and annotation, and copy number variation (CNV) detection. All data were integrated in the dedicated interface Polydiag developed by the bioinformatics platform at the IMAGINE Institute to check coverage of the targeted regions, to sort and filter the called variants by impact and frequency, and to identify relevant candidate mutations and/or CNVs for molecular diagnosis.

Sanger sequencing was performed to validate and segregate the identified TUBB1 (NM_030773) variants (3500xL Genetic Analyzer, Thermo Fisher Scientific, Waltham, MA, USA) with listed primers in Table EV2.

### Burden test

Rare variant burden testing was performed for the TUBB1 gene using the CAST collapsing method (Morgenthaler & Thilly, 2007) in 270 patients with TD, including 193 with ectopia, hemithyroid or hypoplasia and 77 with athyreosis. Contrary to a single-variant association test, variants of the same gene were aggregated and considered as a whole. For each individual, the presence or absence of variants in the gene was noted. The aim of the method was then to count and compare the number of individuals carrying at least one potentially deleterious variant in the TUBB1 gene, in the two groups of individuals. To do so, a likelihood ratio test was performed to compare the TD patients with 406 Caucasian controls from the 1,000 Genomes project phase 3 (The 1,000 Genomes Project Consortium, 2015). Disruptive in-frame, frameshift, missense, splice-acceptor, splice-donor, start-lost, stop-gained or stop-lost variants were considered

deleterious. All deleterious variants with a minor allele frequency < 1% in the ExAC database r0.2 were included in the analysis. We first studied all 270 TD patients then only the 193 TD patients without athyreosis group.

### Human thyroid tissue samples

After approval by our institutional review board of the experimental design and protocols, embryonic thyroid tissue was obtained from products of elective termination of pregnancy and adult thyroid tissues from patients undergoing thyroid surgery.

### Animals

$Tubb1^{-/-}$ mice were previously generated by replacing exons 3 and 4, encoding amino acids 56–451, with a neomycin-resistance gene cassette as previously described (Schwer *et al*, 2001). $Tubb1^{+/-}$ mice on a mixed 129/Sv-BALB/c background were interbred with C57BL/6J mice over ten generations to generate homozygous null mutants ($Tubb1^{-/-}$) with the C57BL/6J background (B6.CG-β1tubulin™). All experiments were conducted in accordance with French regulations and were approved by the Strasbourg Regional Ethics Committee for animal experimentation (C.R.E.M.E.A.S., CEEA 35). Animals were housed in a temperature-controlled room on a 12-h light/12-h dark cycle and had free access to food and water. All adult mice were male. Thyroids at different embryonic stages from E13.5 to E17.5 and adult thyroids at 3 months of age were obtained from wild-type and $Tubb1^{-/-}$ mice and microdissected as described previously (Carre *et al*, 2011).

### Assays on mouse serum samples

Aortic blood samples were collected from 3-month-old wild-type and $Tubb1^{-/-}$ mice. Radioimmunoassays were used to measure serum TSH and serum total T4 after iodothyronine extraction (Dr. S. Refetoff, Chicago, IL, USA) as previously described (Pohlenz *et al*, 1999).

### Flow cytometry of mouse thyroid cells

Mouse thyroid tissue from E17.5 embryos (15 pooled thyroids per sample) and adults (four thyroids per sample) were microdissected, cleansed of fat and connective tissue, and placed in ice-cold phosphate-buffered saline (PBS) containing 2% foetal calf serum (FCS). Cells were prepared and sorted by flow cytometry as previously described (Gawade *et al*, 2016). Briefly, single-cell suspensions were obtained by enzymatic digestion with 1 mg/ml collagenase/dispase and 2 μg/ml DNase I (Roche Diagnostics, Basel, Switzerland) at 37°C for 20 min. The cells were then centrifuged with PBS containing 2% FCS and stained with cell surface markers for 20 min. Finally, the cells were acquired on a BD FACSAria II flow cytometer (Becton Dickinson, Franklin Lakes, NJ, USA). The following monoclonal antibodies were used: EpCAM/CD326 (1:1,000, clone G8.8, # 118216), PDGFRa/CD140a (1:400, APA5, # 135906), CD45 (1:200, clone 30-F11, # 103128) and Pecam/CD31 (1:400, clone 390, # 102406) from BioLegend (San Diego, CA, USA); and CD41 (1:50, clone MWReg30, # 553848, Becton Dickinson). The secondary antibody was goat anti-rabbit A647 from Life Technologies (1:2,000;

Carlsbad, CA, USA). Each pool of sorted cells was collected in RLT buffer from the Qiagen RNeasy MicroKit (Qiagen, Valencia, CA, USA) for RNA extraction experiments.

### RNA extraction and quantitative RT–PCR

The thyroids were microdissected and immediately snap-frozen and stored at −80°C. Total RNA of sorted cells or thyroid tissue was isolated using the Qiagen RNeasy MicroKit or MiniKit (Qiagen). The Maxima First Strand cDNA Synthesis Kit (Thermo Fisher Scientific) was used for reverse transcription of 250 ng of each RNA sample. The synthesized cDNA was diluted to 1/20, and 5 μl was used for each PCR. Each reaction consisted of TaqMan Universal PCR Master Mix or SybrGreen PCR Master Mix (Thermo Fisher Scientific) and primers. Peptidylprolyl isomerase A served as an endogenous control. Real-time PCR was performed using the QuantStudio 3 Real-Time PCR System (Thermo Fisher Scientific). The data were analysed using the comparative cycle threshold method and reported as the fold change in gene expression, normalized for a calibrator of value 1. Primers sequences for human *TUBB1* were as follows: Forward GGGACGATGGACAGCATTCGAT and Reverse ACCTCTAGGACATTCTCGATCAGC. Primers sequences for mice α and β-tubulin are listed in Appendix Fig S4B.

### Immunohistochemistry and quantification

Human or mouse tissues were fixed by immersion in 3.7% buffered formalin then embedded in paraffin. Subsequently, 4-μm-thick sections were mounted on StarFrost adhesive slides (Knittel Glaser, Braunschweig, Germany) and processed for immunohistochemistry, as previously described (Carre *et al*, 2011). The primary antibodies were used at the following dilutions: rabbit antibody to human or mouse β1-tubulin, 1:1,000 (donated by François Lanza), rabbit anti-Ecadherin, 1:100 (# 610682, Becton Dickinson), mouse anti-TG, 1:100 (# M0781, DakoCytomation, Glostrup, Denmark), rabbit anti-Nkx2-1, 1:2,500 (#PA0100, Biopat, Italy), mouse anti-T4, 1:10,000 (clone BGN/0980/322, # 8959-9831, AbD Serotec, Raleigh, NC, USA), rabbit anti-calcitonin, 1:400 (# A0576, DakoCytomation), mouse anti-Ki67, 1:20 (# 550609, Becton Dickinson) and rabbit anti-KDEL, 1:1,500 (# PA1-013, Thermo Fisher Scientific). The fluorescent secondary antibodies were Alexa Fluor 594 goat anti-rabbit and Alexa Fluor 488 goat anti-mouse antibodies (1:400, Thermo Fisher Scientific). The nuclei were stained using the Hoechst 33,342 fluorescent stain (0.3 mg/ml; Thermo Fisher Scientific). Photographs were taken using a fluorescence microscope (Leitz DMRB; Leica, Wetzlar, Germany) and digitized using a chilled 3CCD camera (C5810; Hamamatsu Photonics, Hamamatsu City, Japan).

The sections were then analysed using ImageJ 1.32s (freeware, www.rsbweb.nih.gov/ij) as previously described (Carre *et al*, 2011; Kariyawasam *et al*, 2015). The Nkx2-1-positive surface areas per section allowed us to draw the total thyroid surface area in μm². The surface areas positive for calcitonin and T4, two markers of late thyroid differentiation, were normalized for total thyroid surface area. For stained surface quantification, we used one of every two sections at E9.5 and E11.5, one of every five sections at E13.5 and five sections per adult thyroid (3 months of age). We determined the surface area to obtain an estimate of the total stained surface for each thyroid and each marker. Proliferation of Nkx2-1-positive cells

at E9.5 was estimated by counting Ki67-positive nuclei among Nkx2-1-positive cells on every other section throughout the entire tissue sample at E9.5. At least three thyroids were analysed per genotype. The results are reported as mean ± SEM.

For Nkx2-1 staining of adult mouse thyroid glands, the first immunohistochemistry steps were as described above. After application of the primary antibody, the sections were incubated with biotinylated secondary antibody for 1 h. Immunostaining was then performed using the Vectastain ABC Kit (Vector Laboratories, Burlingame, CA, USA) according to the manufacturer's instructions. The sections were then incubated in 3,3′-diaminobenzidine tetrahydrochloride and counterstained with hemalum–eosin.

### Western blot studies of mouse thyroid tissue

Proteins prepared from mouse thyroid tissue collected in RIPA buffer and sonicated were quantified using the BCA protein assay (Thermo Fisher Scientific). Then, 20 μg of total protein was separated on Bis–Tris polyacrylamide gel with a 4–12% gradient (Thermo Fisher Scientific) and transferred onto PVDF membranes (Thermo Fisher Scientific). The membranes were incubated with the primary antibodies mouse anti-Chop (1:1,000, # 2895, Cell Signaling Technology, Danvers, MA, USA) or rabbit anti-Actin (1:2,000, # A5441, Sigma-Aldrich) antibodies, followed by horseradish peroxidase-conjugated goat anti-mouse or anti-rabbit antibodies. Binding of secondary antibodies was revealed using the Amersham ECL Prime Detection Reagent Kit (GE Healthcare, Chicago, IL, USA). The protein bands on the membranes were scanned with the ImageQuant LAS 4000 Station (GE Healthcare) and then analysed using ImageJ 1.32s to determine the protein levels, with Actin protein serving as an internal control.

### Molecular modelling of the P160L mutated protein

The wild-type human TUBB1 sequence (accession number: Q9H4B7) was downloaded from the UniProt database, and the P160L mutation introduced into it. Both the wild-type and mutant TUBB1 sequences were modelled using Modeller 9.18 software (Šali & Blundell, 1993) with PDB 4I4T chain B as the template (Prota et al, 2013). The models were analysed using PyMOL visualization software (DeLano, 2002).

### Electron microscopy

Samples were fixed for 1 h in 3% glutaraldehyde in PBS buffer, washed and embedded in Epon. 90-nm sections were collected on nickel grids and contrasted with uranyl acetate and lead citrate. Acquisitions were performed with a Gatan Orius 1000 CCD Camera (Gatan, Pleasanton, CA, USA) on a JEOL 1011 transmission electron microscope (JEOL, Tokyo, Japan).

### Plasmids, cell cultures, transfection and immunofluorescence

We used the phumanTUBB1-tagged Myc vector described by Kunishima et al (2009). Mutant P160L-TUBB1 was generated using a PCR-based site-directed mutagenesis method as described previously, using the Stratagene QuikchangeVR Kit (Agilent Technologies; Carré et al, 2007). Nthy (Nthy-ori 3.1; given by Corinne Dupuy)

immortalized human thyroid-cell lines were cultured as previously described and used from passage 12 (Lemoine et al, 1989). The Nthy cells were plated at $0.4 \times 10^5$/well on poly-L-lysine-coated slides in 12-well plates 24 h before transfection then transfected with 500 ng of vectors containing wild-type or P160L mutant TUBB1 using XtremeGENE-HP-DNA, as recommended by the manufacturer (Roche Applied Science, Penzberg, Germany). After 24 h, cells were used for immunofluorescence as already described (Bourg et al, 2015). The cells were washed with pre-warmed PHEM buffer; fixed with 4% PFA, 0.2% glutaraldehyde and 0.5% Triton; and permeabilized with PBS-Triton 0.1%. Immunostaining was performed with rabbit anti-Myc antibody (# 2272, 1:500, Cell Signaling Technology) and mouse anti-α-tubulin (DM1A, # T9026, 1:1,000, Sigma-Aldrich, Saint-Louis, MI, USA) then with Alexa Fluor 647 goat anti-rabbit and Alexa Fluor 555 goat anti-mouse antibodies (1:400, Thermo Fisher Scientific).

### Human megakaryocytes and proplatelet formation

CD34$^+$ cells were isolated from peripheral blood using an immunomagnetic technique according to the manufacturer's instructions (#130-046-70, Miltenyi Biotec, Bergisch Gladbach, Germany). Briefly, 100 μl FcR blocking reagent and 100 μl CD34 MicroBeads were incubated with $10^8$ cells. The remaining population was cultured at 37°C in 5% $CO_2$ in Iscove's modified Dulbecco's medium (IMDM; Thermo Fisher Scientific) supplemented with 15% BIT 9500 serum substitute (Stemcell Technologies, Vancouver, Canada), α-monothioglycerol (Sigma-Aldrich) and liposomes (phosphatidylcholine, cholesterol and oleic acid; all from Sigma-Aldrich), in the presence of human recombinant stem cell factor (SCF, 20 ng/ml, Miltenyi Biotec) and human thrombopoeitin (50 nM, Miltenyi Biotec) added once on day 0 to the culture medium, followed by 20 nM thrombopoeitin alone on day 6 with no further SCF addition. For proplatelet formation assays, megakaryocytes were plated on a BSA-coated surface (chamber slide, Ibidi, Martinsried, Germany) on day 10. On day 13 or 14, the megakaryocytes were fixed using 4% paraformaldehyde and stained for β-tubulin.

### Preparation of washed platelets

To obtain human platelets, venous blood from healthy donors or patients was collected in 10% ACD/A buffer (75 mM sodium citrate, 44 mM citric acid, 136 mM dextrose, pH 4.5). Platelets were washed as previously described (Adam et al, 2003) in the presence of apyrase (100 mU/ml) and prostaglandin E1 (1 μM) to minimize platelet activation. Platelet counts in patients and controls were adjusted to similar levels ($3 \times 10^8$ platelets/ml) in Tyrode's buffer (137 mM NaCl, 2 mM KCl, 0.3 mM $NaH_2PO_4$, 1 mM $MgCl_2$, 5.5 mM glucose, 5 mM N-2-hydroxyethylpiperazine-N′-2-ethanesulfonic acid, 12 mM $NaHCO_3$ and 2 mM $CaCl_2$, pH 7.3).

### Platelet aggregation

Platelet aggregation was monitored by measuring light transmission through a stirred suspension of washed platelets ($3 \times 10^8$/ml) at 37°C using a Chrono-Log Aggregometer (Chrono-Log Corporation, Havertown, PA, USA), as previously described (Adam et al, 2010). Platelet aggregation was triggered by ADP and type I collagen (Chrono-log Corp.).

## Flow cytometry of human platelets

Whole blood from healthy donors or patients was diluted in PBS to obtain a platelet concentration of $2.5 \times 10^7$/ml. Diluted whole blood was then incubated with phycoerythrin (PE)-anti-human CD62P (clone AK-4; eBioscience, Thermo Fisher Scientific, 5 μl anti-CD62P-PE/$5 \times 10^5$ platelets) or fluorescein isothiocyanate (FITC) anti-human-activated $\alpha_{IIb}\beta_3$ integrin (clone PAC-1; #340507, Becton Dickinson, 20 μl PAC-1-FITC/$5 \times 10^5$ platelets) according to manufacturer's instructions for 20 min at room temperature. The samples were then analysed directly with an Accuri C6 flow cytometer (Becton Dickinson).

## Western blotting study of human platelets

Washed platelets (300 μl; $3 \times 10^8$/ml) were lysed in Laemmli sample buffer (10 mM HEPES, 2% SDS, 10% glycerol and 5 mM EDTA). The proteins were separated by sodium dodecyl sulphate (SDS)-polyacrylamide gel electrophoresis and transferred to nitrocellulose membranes, which were incubated with the primary antibodies rabbit anti-α tubulin (1:1,000, Clone EP1332Y; Merck Millipore, Billerica, MA, USA) or rabbit anti-β1-tubulin antibody (1:1,000, donated by François Lanza). Immunoreactive bands were visualized with enhanced chemiluminescence detection reagents (Perbio Science, Thermo Fisher Scientific) using a G:BOX Chemi XT16 Image System and then quantified using Gene Tools version 4.03.05.0 (Syngene, Cambridge, UK).

## Statistics

Sample size determination was based on previous experience with similar studies. Results are reported as mean ± SEM for the number of experiments indicated in the figure legends. Statistical analyses were performed using GraphPad Prism4 (GraphPad, La Jolla, CA, USA). Data were analysed by one-way ANOVA followed by Dunnett's test, except for those parameters involving comparison of only two experimental groups, in which case an unpaired Student's *t*-test was used as indicated in the figure legends. Differences were considered significant when $P < 0.05$. All *P*-values for figures can be found in Appendix Table S3.

# Data availability

The datasets produced in this study are available in the following database: Clinical data: ClinVar accession numbers: SCV000840553.1, SCV000840554.1 and SCV000840555.1 (https://www.ncbi.nlm.nih.gov/clinvar/?term=SCV000840553.1, http://www.ncbi.nlm.nih.gov/clinvar/?term=SCV000840554.1 and http://www.ncbi.nlm.nih.gov/clinvar/?term=SCV000840555.1).

**Expanded View** for this article is available online.

## Acknowledgements

This study was funded by the French public non-profit funding agency *Programme Hospitalier de Recherche Clinique* (ClinicalTrial.gov: NCT01916018, P 110120 – IDRCB 2012-A00797-36). AC, and MP received financial support from three corporations (EDF, Sandoz SAS, and Merck Serono France) and from the non-profit Princess Grace Foundation of Monaco. AC was funded by SFEDP (Société Française d'Endocrinologie et Diabétologie Pédiatrique). AS was supported by the European Society for Paediatric Endocrinology Research Fellowship Grant and Alexander S. Onassis Foundation. CJ has received support under the program "Investissements d'Avenir" launched by the French Government and implemented by ANR with the references ANR-10-LBX-0038, ANR-10-IDEX-0001-02 PSL. KN was supported by the FRM fellowship SPF20140129173. We thank the patients and families for their participation, the URC-CIC Paris Centre (Sandra Colas and Emilie Ervilus) for study implementation and monitoring, and the IMAGINE Institute Biobank. We are deeply grateful to Beatrice Durel and Pierre Bourdoncle at the Cochin Imaging Facility. Our gratitude extends also to Monique Freund and her team at the animal facility in Strasbourg for technical assistance with the animal studies. We thank Prof. Samuel Refetoff (University of Chicago) for performing the plasma hormonal assays; and Daphné Geloen (Necker Children's Hospital, Biological Haematology Department, Paris) and Christelle Repérant (U1176 INSERM, Le Kremlin-Bicêtre) for their technical assistance with the platelet studies.

## Author contributions

AC and MP coordinated and instigated the study with FA and DB. ASt, DK, JL and MP provided clinical samples and data. AC, DK, ASt, CB, AMu and MP

---

### The paper explained

**Problem**

Congenital hypothyroidism is the most common endocrine neonatal disorder with an estimated prevalence of 1/3,500 newborns. It is mainly caused by defects in thyroid development (thyroid dysgenesis) or thyroid hormone synthesis. Among patients with congenital hypothyroidism due to thyroid dysgenesis, the specific genetic cause is identified only in < 5% of patients. Identifying the molecular defect allows early clinical care of hypothyroidism and associated malformations, and provides new insights into thyroid development and disease.

**Results**

Using whole exome sequencing, we uncovered an homozygous missense mutation in *TUBB1* gene, in two siblings of a consanguineous family with congenital hypothyroidism and thyroid dysgenesis (TD). By direct *TUBB1* sequencing in a cohort of 270 patients with TD, we identified two more mutations in two distinct families with TD. *TUBB1* gene encodes for a β1-tubulin, and until now, reported roles for this protein were confined to platelets. Our functional studies show TUBB1 expression in the developing and adult thyroid in humans and mice. The $Tubb1^{-/-}$ mice have large platelets and show hypothyroidism, in accordance with the phenotype in mutated patients. Thyroids of $Tubb1^{-/-}$ exhibited proliferation defects during early development (embryonic day E9.5), altered migration at E11.5 and E13.5, and failure of hormone secretion at E17.5 and adulthood. All these mechanisms require proper microtubule function. Interestingly, two of the novel *TUBB1* mutations were associated with basal activation and exaggerated aggregation of platelets.

**Impact**

This is the first time that *TUBB1* mutations are associated with thyroid dysgenesis, in addition to abnormal platelet physiology. These findings expand the spectrum of the rare paediatric diseases related to tubulin mutations and provide new insights into the genetic background and mechanisms involved in congenital hypothyroidism. A *TUBB1* mutation screening study in patients with non-autoimmune, non-postsurgical hypothyroidism and altered mean platelet volume and/or a history of thrombotic disease should be considered.

planned the genetic work. AC and CB performed exome sequencing data. AC performed molecular studies and imaging data. GS and SG performed flow cytometry studies. FJ-H and FT performed Burden test. AC, FA, DB and MP analysed the data. CS and FL provided mice and performed mouse platelets studies. FA, AK, DL and DB performed human platelets studies. KN performed the modelling studies. CB-F coordinated the WES procedure. PN gave bioinformatics support. ASc and AMi performed electron microscopy studies. AC, ASt, FA and MP draft and finalized the manuscript with the help of DB, CJ, FL, and RS.

## Conflict of interest

The authors declare that they have no conflict of interest.

## For more information

(i)   ExAC Browser, http://exac.broadinstitute.org
(ii)  1000 Genomes, http://www.1000genomes.org
(iii) dbSNP, http://www.ncbi.nlm.nih.gov/projects/SNP/
(iv)  PolyPhen-2, http://genetics.bwh.harvard.edu/pph2/
(v)   SIFT, http://sift.bii.a-star.edu.sg

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
