## [Review Process File · EMBO Molecular Medicine]

***TUBB1* mutations cause thyroid dysgenesis associated with abnormal platelet physiology**

Athanasia Stoupa, Frédéric Adam, Dulanjalee Kariyawasam, Catherine Strassel, Sanjay Gawade, Gabor Szinnai, Alexandre Kauskot, Dominique Lasne, Carsten Janke, Kathiresan Natarajan, Alain Schmitt, Christine Bole-Feysot, Patrick Nitschke, Juliane Léger, Fabienne Jabot-Hanin, Frédéric Tores, Anita Michel, Arnold Munnich, Claude Besmond, Raphaël Scharfmann, François Lanza, Delphine Borge, Michel Polak, and Aurore Carré

Review timeline:

Submission date:	19 July 2018
Editorial Decision:	21 August 2018
Revision received:	11 September 2018
Editorial Decision:	11 October 2018
Revision received:	18 October 2018
Accepted:	19 October 2018

Editor: Céline Carret

Transaction Report:

1st Editorial Decision

21 August 2018

Thank you for the submission of your manuscript to EMBO Molecular Medicine. We have now heard back from the two referees whom we asked to evaluate your manuscript.

You will see from the set of comments pasted below that while both referees found the study to be important and novel, they both provide a long list of questions, suggestions, and requests that we would like to encourage you to address in a next final version of your article to improve conclusiveness and clarity. Please note that EMBO Molecular Medicine strongly supports a single round of revision and that, as acceptance or rejection of the manuscript will depend on another round of review, your responses should be as complete as possible.

I look forward to receiving your revised manuscript.

***** Reviewer's comments *****

Referee #1 (Remarks for Author):

This paper describes three different loss-of-function mutations in TUBB1 in patients with thyroid dysgenesis (TD) associated with abnormal platelet biology and morphology. It starts with a consanguineous pedigree (family 1) in which two sisters with congenital hypothyroidism (CH) due to ectopy are homozygous for a missense mutation in TUBB1. However, a brother who is homozygous for the same mutation has neither CH nor ectopy. A search for TUBB1 mutations in 270 patients with CH due to TD (CHTD) revealed a heterozygous mutation in two girls with severe CH and ectopy and in their unaffected fathers. In a paternal aunt in family 3 who carries the mutation, there was mild hypothyroidism and a right hemithyroid. Platelet counts were normal but platelet volume tended to be high. The only mutation (P160L) that should give rise to a full-length protein was shown in the human thyroid cell line Nthy-ori to decrease incorporation of the protein into microtubules.

The authors further demonstrate that:

- a) Beta1-tubulin mRNA and protein are expressed in mouse and human thyroids.
- b) K-O mice have abnormal thyroid migration (see comment about figure Fig 3a below) during embryogenesis and a higher serum TSH and a lower serum T4 in adulthood.
- c) Platelet activation and aggregation were abnormal in some of the humans with mono -or biallelic TUBB1 mutations.

The findings are novel and a huge amount of work is presented. In fairness, the authors should acknowledge that the evidence for a purely Mendelian transmission of CHTD is limited and that it likely results from two hits: a germline predisposing mutation (accounting for the 40-fold higher relative risk of CHTD in the first degree relatives of an affected individual) coupled with a somatic event (accounting for the almost universal discordance between MZ twins). Another mystery in ectopy is the three to one female-to-male ratio, and it is striking that the four patients with ectopy in the present study are females. The numbers are small but a comment is in order. Lastly, heterozygous mutations were seen in severe CH with ectopy (P6) and in individuals with normal thyroid migration and function (P7), which may fit with random autosomal monoallelic expression in the thyroid.

Specific comments:

- page 4, line 3: in the Deladoey reference, the overall prevalence of CH was one in 2,500, while that of CH due to TD was one in 4,270. In the Barry reference, the prevalence of CH due to TD was about one in 5,000. Please clarify.
- page 4, line 17: 5% here and 10 % on page 33, line 6. The former is probably already excessive.
- page 4, line 20: childhood-onset TD is a NON-SEQUITUR - the only thyroid development that occurs after birth is growth, not differentiation or migration.
- page 6, line 7 and page 11, line 21: with a norm of 7 ± 3 mL, a volume of 3.1 mL is -1.3 SD, i.e. within the normal range. How many normal people have a pyramidal lobe?
- page 7, line 8: left hemi-agenesis should be right hemithyroid (i.e, there may have been a left lobe during early embryogenesis which later involuted -as does the thyroid of NKX2.1 -/- mice embryos).
- page 7, line 16: the asymmetry is indeed mild and can likely be seen in the general population.
- page 11, line 3: Figure S4a, not S4b.
- page 12, line 12: increases in INTRATHYROIDAL T4 and CT.
- page 20, line 10: the qualifier 'with variable penetrance, is essential here, since some carriers of the TUBB1 mutation (even biallelic) had no thyroid phenotype at all, emphasizing the fact that this germline mutation may be necessary to develop CHTD but is clearly not sufficient.
- page 20, line 19: along the same line, replace 'causative, by 'predisposing'.

Figure and tables:

- Show pedigree on a separate figure with data from Table 1 under each symbol and abnormal values in bold (see papers by Refetoff's group).
- Fig 3a: the concept of a migration delay is based on a very slight difference between WT and Tubb1-/- at E11.5 and does not affect all cells in the latter. This may be reminiscent of the 'dual

ectopy' which is seen in 9 % of newborns with CH due to ectopy (1). This said, it would be more convincing to show that the image at E11.5 in K-O mice is similar to that in WT at, say, 10.5.
-Fig 3c: units for y-axis not labeled.

1. Wildi-Runge S, Stoppa-Vaucher S, Lambert R, Turpin S, Van Vliet G, Deladoey J 2012 A High Prevalence of Dual Thyroid Ectopy in Congenital Hypothyroidism: Evidence for Insufficient Signaling Gradients during Embryonic Thyroid Migration or for the Polyclonal Nature of the Thyroid Gland? *J Clin Endocrinol Metab* 97:E978-E981.

Referee #2 (Comments on Novelty/Model System for Author):

The study is well implemented and the novelty of discovering a new gene for thyroid dysgenesis is high. Genetic panels for this disease will be expanded as a result, and there is also an interesting overlap with platelet function that will lead to further clinical studies.

Referee #2 (Remarks for Author):

This manuscript reports a very well executed study that identified a new genetic cause for a rare congenital form of hypothyroidism. The human genetic evidence is solid and it is very well complemented not only by *in vitro* studies but also by a mouse knockout model. Even though there are some issues that should be addressed to improve the manuscript, they do not detract from its novelty and importance.

Major comments

1. The Introduction is not well written. It is essentially an extended abstract, going beyond the background and rationale of the study to present and discuss the results. It also includes inaccurate and misleading statements, such as the one on the study's objective (page 5, lines 14-15), which, if the study is described as it was really done, was NOT to "further assess links between TUBB1 mutations and TD in humans", but to find new genes for TD. The Introduction should be re-written, and most of the text should be removed (especially the second paragraph).
2. Page 6, lines 8-10: I find it very puzzling that the parents consented to genetic testing but "were not able to undergo thyroid ultrasound".
3. Page 7, lines 4-5: Same comment. The phenotyping is incomplete and the paper should acknowledge that more explicitly.
4. Page 7, lines 15-16: Here, the difference in volume between the two thyroid lobes is taken as evidence of a partial disease phenotype, and the person is coded as affected, despite normal thyroid function, solely based on the presence of this difference. This is difficult to justify, because there are no objective cut-offs to assess whether the difference is normal or abnormal. Moreover, since thyroid lobe volumes are calculated, they are prone to measurement errors. The status of this person should be more accurately coded as unknown (question mark).
5. Page 7, line 25: Is there a max. possible CADD score? If so, provide it, to put the cited scores into perspective.
6. Page 8: The burden test is not well described, neither in the Results nor in the Methods. Include a supplement where all the variants analyzed are listed, including any known disease associations for specific variants.
7. Page 8, lines 7 and 10: In addition to the p-values, give also the absolute numbers and percentages of subjects with variants in the TD and control groups.
8. Page 8, line 12 and Figure 1b: Expand the alignment by including sequences from invertebrates (*Drosophila* and *C. elegans*).

9. Figure 1c: In the graph, show the identity and location of mutations associated with congenital macrothrombocytopenia.
 10. Discussion: The paper discusses the suggestion to screen patients with TD and coagulation problems for TUBB1 mutations. It should also discuss more explicitly the suggestion to assess thyroid function and morphology in patients with congenital macrothrombocytopenia
 11. Page 12, line 12: The difference in calcitonin was not statistically significant. Do not claim a difference then. Rephrase.
 12. Page 20, line 14: and "INCOMPLETE PENETRANCE WITH variable expressivity for TD"
 13. Page 20, line 10: "A model of dominant inheritance". This is OK but could be discussed better, in the light also of the comment above, the small % of TD patients with TUBB1 mutations, the many genes already known to be associated with TD, and the lack of a known genetic cause in the vast majority of cases. Diseases with these characteristics are often oligogenic in nature. The authors could discuss this and cite other endocrine-related diseases as examples (e.g., congenital hypogonadotropic hypogonadism, or Bardet-Biedl syndrome).
 14. Page 21, Patients: Some more information should be given about the cohort.
 15. Page 31, Statistics: The use of statistical methods is inconsistent in the paper. Most comparisons that include multiple groups are analyzed by ANOVA, except those in Figure 2, which are done by multiple t-tests. Those should then be redone by ANOVA.
 16. Figure 2c: Why is there no variation in the right EpCam bar?
 17. Table 1, Family 2, ExAC frequency: If the mutation is not listed in ExAC, I think one can say that its frequency in ExAC is ZERO.
 18. Name the ethics committee that approved the human studies.
- Other comments
19. Remove « unexpected » from the title. It does really not add anything.
 20. Page 3, line 7: remove "in vivo", it is redundant.
 21. Page 3, line 13: replace "tubulin mutations" with "mutations in tubulin-coding genes"
 22. Page 4, line 1: INTRODUCTION (capitals)
 23. Page 4, line 21: Move the references after WES in line 19, otherwise the reader gets the impression that this mutation was identified in previously published work.
 24. Page 4, line 25: "that heterodimerize TO FORM a/b tubulin"
 25. Page 5, line 23: remove "(F1)" and "thyroid dysgenesis" (use TD, which has already been spelled-out).
 26. Page 8, line 18: "consisting in" should be changed to "manifesting as", or something similar.
 27. Page 9, line 2: "with stringent sorting regions". What does this mean? Explain in the text.
 28. Page 9, line 5: "and IN adulthood"
 29. Page 10, line 9: Explain that these were Myc-tagged, not native proteins.
 30. Page 10, line 13: "defects AT the structural level"

31. Page 10, line 15: Qualify the statement: "LIKELY lead to..."
32. Page 10, line 21: Give here also the reference for the mice.
33. Page 10, line 22: "As expected". This is explained only later, so either give a reference here, or remove it.
34. Page 12, line 25: "versus" should be "compared to". "corroborate with" should be "consistent with".
35. Page 13, line 7: "these data indicated PARTIAL thyroid tissue disorganization".
36. Page 13, line 13: "Analysis of platelets FROM patients..." "Human" is redundant.
37. Page 16, line 2: "were also seen".
38. Page 14, lines 3 and 4: Remove "the".
39. Page 16, line 10: "contrasting with" should be replaced by "but".
40. Page 16, line 20: Remove "the".
41. Page 18, lines 18-19: The two references cited are both from the 1970s, so either delete "and 1980s" or give a reference from the 1980s.
42. Page 18, line 25: "presence of ER stress markers" is not very meaningful. Rephrase to "increased expression of ER stress markers" or something similar.
43. Page 19, line 8: microtubuleS
44. Page 19, last paragraph: The reader gets the impression that there are only 3-4 TUBB1 mutations known to be associated with congenital macrothrombocytopenia. If so, say it explicitly. If not, rephrase accordingly.
45. Page 20, line 1: After a thrombotic episode, anti-platelet drugs are a standard of care and would be given anyhow. Delete "treatment for and/or".
46. Page 20, line 17: The % of TD patients with TUBB1 mutations should be given also in the Results and in the Abstract.
47. Page 25, line 15: In the paper, the control gene is mentioned as "peptidyl propyl isomerase 1", not "peptidylpropyl isomerase A". Harmonize.
48. For all antibodies, provide the supplier's catalogue number. For the donated antibody, if a catalogue number is not available, provide a publication as reference.
49. Page 28, line 12. Give the source of the Nthy cell line (Sigma?) and the passage number (or range) when it was used for experiments.
50. Legend of Figure 1: Explain in the legend what the "?" sign means.
51. Page 33, Problem. For clarity and accuracy, rephrase to: "Among patients with congenital hypothyroidism due to thyroid dysgenesis, the specific genetic cause is identified only in less than 10% of patients."
52. Page 34, line 3: Add: "in patients with NON-AUTOIMMUNE, NON-POSTSURGICAL hypothyroidism".
53. Page 41, lines 23-24: "normalized to one of three thyroid tissues"

54. Page 42, line 4: as above.

55. Page 42, line 4: normalized TO.

56. Page 44, line 1: Change HAD to HAVE.

57. Table 1: I think "Maghreb" covers both Algerian and Moroccan, which are likely not different from one another.

1st Revision - authors' response

11 September 2018

Referee #1 (Remarks for Author):

This paper describes three different loss-of-function mutations in TUBB1 in patients with thyroid dysgenesis (TD) associated with abnormal platelet biology and morphology. It starts with a consanguineous pedigree (family 1) in which two sisters with congenital hypothyroidism (CH) due to ectopy are homozygous for a missense mutation in TUBB1. However, a brother who is homozygous for the same mutation has neither CH nor ectopy. A search for TUBB1 mutations in 270 patients with CH due to TD (CHTD) revealed a heterozygous mutation in two girls with severe CH and ectopy and in their unaffected fathers. In a paternal aunt in family 3 who carries the mutation, there was mild hypothyroidism and a right hemithyroid. Platelet counts were normal but platelet volume tended to be high. The only mutation (P160L) that should give rise to a full-length protein was shown in the human thyroid cell line Nthy-ori to decrease incorporation of the protein into microtubules.

The authors further demonstrate that:

- a) Beta1-tubulin mRNA and protein are expressed in mouse and human thyroids.
- b) K-O mice have abnormal thyroid migration (see comment about figure Fig 3a below) during embryogenesis and a higher serum TSH and a lower serum T4 in adulthood.
- c) Platelet activation and aggregation were abnormal in some of the humans with mono -or biallelic TUBB1 mutations.

The findings are novel and a huge amount of work is presented. In fairness, the authors should acknowledge that the evidence for a purely Mendelian transmission of CHTD is limited and that it likely results from two hits: a germline predisposing mutation (accounting for the 40-fold higher relative risk of CHTD in the first degree relatives of an affected individual) coupled with a somatic event (accounting for the almost universal discordance between MZ twins). Another mystery in ectopy is the three to one female-to-male ratio, and it is striking that the four patients with ectopy in the present study are females. The numbers are small but a comment is in order. Lastly, heterozygous mutations were seen in severe CH with ectopy (P6) and in individuals with normal thyroid migration and function (P7), which may fit with random autosomal monoallelic expression in the thyroid.

Firstly, we would like to thank you for your positive feedback and your helpful comments which we have incorporated in our paper.

Specific comments:

-page 4, line 3: in the Deladoey reference, the overall prevalence of CH was one in 2,500, while that of H due to TD was one in 4,270. In the Barry reference, the prevalence of CH due to TD was about one in 5,000. Please clarify.

This has been clarified in the text as follows: "the most common neonatal endocrine disorder affecting one in 2500-3500 newborns (Barry *et al*, 2016, Deladoëy *et al*, 2011). In France, the prevalence of CH due to TD is estimated in 1/5000 (Barry *et al*, 2016)." page 4 line 4

-page 4, line 17: 5% here and 10 % on page 33, line 6. The former is probably already excessive. We corrected by 5% throughout the text.

-page 4, line 20: childhood-onset TD is a NON-SEQUITUR - the only thyroid development that occurs after birth is growth, not differentiation or migration.

We corrected in the text as follows: "CH" instead of childhood-onset TD, page 4 line 21.

-page 6, line 7 and page 11, line 21: with a norm of 7 ± 3 mL, a volume of 3.1 mL is -1.3 SD, i.e. within the normal range. How many normal people have a pyramidal lobe?

Thank you for your comment. The prevalence of pyramidal lobe varies in different studies from 15 to 75% based on different cohort study number and methods of detection (neck and thyroid ultrasound, CT scan, surgery) (Levy et al, 1982, Kim et al, 2013, Mortensen et al, 2014). Data on normal people (general population) are scarce (estimated prevalence of 17%, Levy et al, 1982). Most studies show a predominant position on the left thyroid lobe and a great difference in gender prevalence. Interestingly, in our patient the pyramidal lobe is situated on the right thyroid lobe. This has been added in the text, page 6 line 2.

We have also observed pyramidal lobe in knock-out mice at E15.5 and E17.5, (page 11, line 20 and Appendix Table S2) reinforcing the causative role of *TUBB1* mutations in TD.

Levy HA, Sziklas JJ, Rosenberg RJ, Spencer RP. Incidence of a pyramidal lobe on thyroid scans. Clin Nucl Med. 1982 Dec;7(12):560-1.

Kim DW, Jung SL, Baek JH, Kim J, Ryu JH, Na DG, Park SW, Kim JH, Sung JY, Lee Y, Rho MH. The prevalence and features of thyroid pyramidal lobe, accessory thyroid, and ectopic thyroid as assessed by computed tomography: a multicenter study. Thyroid. 2013 Jan;23(1):84-91.

Mortensen C, Lockyer H, Loveday E. The incidence and morphological features of pyramidal lobe on thyroid ultrasound. Ultrasound. 2014 Nov;22(4):192-8.

-page 7, line 8: left hemi-agenesis should be right hemithyroid (i.e, there may have been a left lobe during early embryogenesis which later involuted -as does the thyroid of NKX2.1 -/- mice embryos).

We have corrected in the text throughout the manuscript.

-page 7, line 16: the asymmetry is indeed mild and can likely be seen in the general population. Indeed the difference in measurements of two thyroid lobes is mild, and likely not significant. The data in literature about size between the two thyroid lobes in general population are rare (Spencer et al, 1965) and suggest that the right lobe is slightly larger than the left (as described in our case). The absence of solid cut-offs in thyroid lobe values in normal people makes it difficult to define P7 as an affected person. We have therefore modified the status of P7 in the familial pedigree F3 with a symbol * signifying "mild thyroid asymmetry with normal thyroid function" (added in the Figure legend page 42 lines 9-10).

Spencer Rp, Waldman R. Size And Positional Relationships Between Thyroid Lobes In The Adult As Determined By Scintillation Scanning. J Nucl Med. 1965 Jan;6:53-8.

-page 11, line 3: Figure S4a, not S4b.

We corrected in the text.

-page 12, line 12: increases in INTRATHYROIDAL T4 and CT.

We added in the text.

-page 20, line 10: the qualifier 'with variable penetrance, is essential here, since some carriers of the *TUBB1* mutation (even biallelic) had no thyroid phenotype at all, emphasizing the fact that this germline mutation may be necessary to develop CHTD but is clearly not sufficient.

Thank you for your comment. We have clarified this in the text: "Indeed, in the described familial pedigrees, some carriers have mild or no thyroid phenotype, suggesting that the *TUBB1* germline mutation may be necessary to be affected by CHTD but it is probably not sufficient to display the phenotype. A second hit (such as a somatic mutation in the thyroid or an epigenetic defect) could be the additional prerequisite to express the disease. Furthermore, the hypothesis of random autosomal monoallelic expression in the thyroid could explain the difference in intrafamilial phenotypic variability in F3. These hypotheses are already documented in the literature for TD (Deladoëy et al, 2007; Magne et al, 2016)".

Page 20 line 25 and page 21 lines 1-7.

-page 20, line 19: along the same line, replace 'causative, by 'predisposing'.
We corrected in the text.

Figure and tables:

-Show pedigree on a separate figure with data from Table 1 under each symbol and abnormal values in bold (see papers by Refetoff's group).

We have modified the Figure1 incorporating the table1 and we have changed the numbering of following figures.

-Fig 3a: the concept of a migration delay is based on a very slight difference between WT and *Tubb1*^{-/-} at E11.5 and does not affect all cells in the latter. This may be reminiscent of the 'dual ectopy' which is seen in 9 % of newborns with CH due to ectopy (1). This said, it would be more convincing to show that the image at E11.5 in K-O mice is similar to that in WT at, say, 10.5.
We agree with your comment. We have observed a mild difference in thyroid migration in KO mice embryos at E11.5 during our experiments. Unfortunately we do not dispose an image corresponding to E10.5. However, this image of tract of migrating cells corresponds to the accurate description of thyroid development published recently by the group of Nilsson and Fagman (Nilsson, Fagman, 2017).

Moreover, we have added your comment in the text page 18 lines 19-20: "The delay of thyroid bud migration could be reminiscent of the dual ectopy seen in 9% of CH due to ectopy (Wildi-Runge et al. 2012)"

Nilsson M, Fagman H. Development of the thyroid gland. *Development*. 2017 Jun 15;144(12):2123-2140

-Fig 3c: units for y-axis not labeled.

We added units for y-axis in the Fig 4c (replaces 3c).

1. Wildi-Runge S, Stoppa-Vaucher S, Lambert R, Turpin S, Van Vliet G, Deladoey J 2012 A High Prevalence of Dual Thyroid Ectopy in Congenital Hypothyroidism: Evidence for Insufficient Signaling Gradients during Embryonic Thyroid Migration or for the Polyclonal Nature of the Thyroid Gland? *J Clin Endocrinol Metab* 97:E978-E981.

Referee #2 (Comments on Novelty/Model System for Author):

The study is well implemented and the novelty of discovering a new gene for thyroid dysgenesis is high. Genetic panels for this disease will be expanded as a result, and there is also an interesting overlap with platelet function that will lead to further clinical studies.

Referee #2 (Remarks for Author):

This manuscript report a very well executed study that identified a new genetic cause for a rare congenital form of hypothyroidism. The human genetic evidence is solid and it is very well complemented not only by in vitro studies but also by a mouse knockout model. Even though there are some issues that should be addressed to improve the manuscript, they do not detract from its novelty and importance.

We would like to thank you for your overall positive assessment of our work.

Major comments

1. The Introduction is not well written. It is essentially an extended abstract, going beyond the background and rationale of the study to present and discuss the results. It also includes inaccurate and misleading statements, such as the one on the study's objective (page 5, lines 14-15), which, if the study is described as it was really done, was NOT to "further assess links between *TUBB1* mutations and TD in humans", but to find new genes for TD. The Introduction should be re-written, and most of the text should be removed (especially the second paragraph).

We have modified the introduction according your comments: pages 4-5.

2. Page 6, lines 8-10: I find it very puzzling that the parents consented to genetic testing but "were not able to undergo thyroid ultrasound".

We recognize that the phenotype of the parents is partly missing but they did not consent to thyroid ultrasound despite their consent for genetic studies.

3. Page 7, lines 4-5: Same comment. The phenotyping is incomplete and the paper should acknowledge that more explicitly.

We agree with your comment; the subject unfortunately did not desire to perform a thyroid ultrasound. This has been added to the text: page 6, lines 22-23 "father carried the same heterozygous mutation; unfortunately thyroid ultrasound was not performed, and complete phenotype was therefore not possible."

4. Page 7, lines 15-16: Here, the difference in volume between the two thyroid lobes is taken as evidence of a partial disease phenotype, and the person is coded as affected, despite normal thyroid function, solely based on the presence of this difference. This is difficult to justify, because there are no objective cut-offs to assess whether the difference is normal or abnormal. Moreover, since thyroid lobe volumes are calculated, they are prone to measurement errors. The status of this person should be more accurately coded as unknown (question mark).

Indeed the difference in measurements of two thyroid lobes is mild, and likely not significant. The data in literature about size between the two thyroid lobes in general population are rare (see reference below Spencer et al. 1965) and suggest that the right lobe is slightly larger than the left (as described in our case). The absence of solid cut-offs in thyroid lobe values in normal people makes it difficult to define P7 as an affected person. We have therefore modified the status of P7 in the familial pedigree F3 with a symbol * signifying "mild thyroid asymmetry with normal thyroid function" (added in the Figure legend page 42 lines 9-10).

Spencer Rp, Waldman R. Size And Positional Relationships Between Thyroid Lobes In The Adult As Determined By Scintillation Scanning. J Nucl Med. 1965 Jan;6:53-8.

5. Page 7, line 25: Is there a max. possible CADD score? If so, provide it, to put the cited scores into perspective.

The PHRED-scaled CADD score is based on the scoring and ranking of all possible single nucleotide variations of the human genomes ($8.6 \cdot 10^9$ SNVs). The score corresponds to the formula - $10 \log_{10}(\text{rank of the variant} / \text{total number of variants})$. So, the score extends from 1 to 99 and a score > 30 means that the variant is among the top 0.1% of the most deleterious variants of the human genome.

6. Page 8: The burden test is not well described, neither in the Results nor in the Methods. Include a supplement where all the variants analyzed are listed, including any known disease associations for specific variants.

The paragraphs of the Methods (page 24 lines 13-18) and the Results (page 7 lines 22-25 and page 8 lines 1-6) have been re-written and a supplementary table has been added (Table EV1).

7. Page 8, lines 7 and 10: In addition to the p-values, give also the absolute numbers and percentages of subjects with variants in the TD and control groups.

The numbers have been added in the re-written text as mentioned above.

8. Page 8, line 12 and Figure 1b: Expand the alignment by including sequences from invertebrates (Drosophila and C. elegans).

Tubb1 sequence in Drosophila and C.elegans is not available in databases, because of the absence of platelets in these species; this is the reason why we did not include this information in the species alignment.

9. Figure 1c: In the graph, show the identity and location of mutations associated with congenital macrothrombocytopenia.

We have added the published mutations (shown in italics) associated with congenital macrothrombocytopenia in the Figure 2B. Please note that taken into account the referees' remarks, we have changed the numbering of figures.

10. Discussion: The paper discusses the suggestion to screen patients with TD and coagulation

problems for TUBB1 mutations. It should also discuss more explicitly the suggestion to assess thyroid function and morphology in patients with congenital macrothrombocytopenia
 We added these sentence page 20 lines 9-10: "On the other side, thyroid function and morphology assessment could be suggested in patients with congenital macrothrombocytopenia."

11. Page 12, line 12: The difference in calcitonin was not statistically significant. Do not claim a difference then. Rephrase.

We corrected in the text as follows page 12 lines 13-16: "Calcitonin-positive thyroid surface area relative to total thyroid surface area tended to be greater in the mutants. Thus, final thyroid differentiation was abnormal in *Tubb1*^{-/-} embryos, with increases in intrathyroidal T4 that probably reflected impaired hormone secretion."

12. Page 20, line 14: and "INCOMPLETE PENETRANCE WITH variable expressivity for TD"
 We corrected in the text.

13. Page 20, line 10: "A model of dominant inheritance". This is OK but could be discussed better, in the light also of the comment above, the small % of TD patients with TUBB1 mutations, the many genes already known to be associated with TD, and the lack of a known genetic cause in the vast majority of causes. Diseases with these characteristics are often oligogenic in nature. The authors could discuss this and cite other endocrine-related diseases as examples (e.g., congenital hypogonadotropic hypogonadism, or Bardet-Biedl syndrome).

Thank you for your comment, we have incorporated these suggestions at the discussion section and enriched this part of discussion, page 20 line 25 and page 21 lines 1-12 as follows: "Indeed, in the described familial pedigrees, some carriers have mild or no thyroid phenotype, suggesting that the TUBB1 germline mutation may be necessary to be affected by CHTD but it is probably not sufficient to display the phenotype. A second hit (such as a somatic mutation in the thyroid or an epigenetic defect) could be the additional prerequisite to express the disease. Furthermore, the hypothesis of random autosomal monoallelic expression in the thyroid could explain the difference in intrafamilial phenotypic variability in F3. These hypotheses are already documented in the literature for TD (Deladoëy et al, 2007; Magne et al, 2016). Finally, the genetics of TD remains complex with mutations in more than 9 known genes and both classical and complex modes of inheritance, such as a suggested oligogenic model by Persani et al (de Filippis et al, 2017). The same genetic pattern of inheritance is also observed in other endocrine-related disorders such as congenital hypogonadotropic hypogonadism (Boehm et al, 2015) or in the more complicated genetic model of Bardet-Biedl syndrome (Muller et al. 2010)."

14. Page 21, Patients: Some more information should be given about the cohort.

2nd Editorial Decision

11 October 2018

Thank you for the submission of your revised manuscript to EMBO Molecular Medicine. We have now received the enclosed reports from the referees that were asked to re-assess it. As you will see the reviewers are now globally supportive and I am pleased to inform you that we will be able to accept your manuscript pending the minor text changes commented by referee 2, and minor editorial amendments.

Corresponding Author Name: Aurore CARRE
Journal Submitted to: EMBO Molecular Medicine
Manuscript Number: EMM-2018-09569